# Why the brown ghost chirps at night

Livio Oboti[1]*, Federico Pedraja[2], Marie Ritter[1], Marlena Lohse[1], Lennart Klette[1], Rüdiger Krahe[1]

[1]Institut für Biologie, Humboldt Universität zu Berlin, Berlin, Germany; [2]Department of Neuroscience, Zuckerman Mind Brain Behavior Institute, Columbia University, New York, United States

## eLife Assessment

This study addresses a significant question in sensory ethology and active sensing in particular. It links the production of a specific signal - electrosensory chirps - to various contexts and conditions to propose that chirps may also serve an active sensing role in addition to their more well-known role in communication. The evidence supporting the role for active sensing is strong. In particular, the evidence showing increased chirping in more cluttered environments and the relationship between chirping and movement are **convincing**. The study provides a lot of **valuable** data, and is likely to stimulate follow-up behavioral and physiological studies.

**\*For correspondence:**
livio.oboti@gmail.com

**Competing interest:** The authors declare that no competing interests exist.

**Abstract** Since the pioneering work by Moeller, Szabo, and Bullock, weakly electric fish have served as a valuable model for investigating spatial and social cognitive abilities in a vertebrate taxon usually less accessible than mammals or other terrestrial vertebrates. These fish, through their electric organ, generate low-intensity electric fields to navigate and interact with conspecifics, even in complete darkness. The brown ghost knifefish is appealing as a study subject due to a rich electric 'vocabulary', made by individually variable and sex-specific electric signals. These are mainly characterized by brief frequency modulations of the oscillating dipole moment continuously generated by their electric organ, and are known as chirps. Different types of chirps are believed to convey specific and behaviorally salient information, serving as behavioral readouts for different internal states during behavioral observations. Despite the success of this model in neuroethology over the past seven decades, the code to decipher their electric communication remains unknown. To this aim, in this study we re-evaluate the correlations between signals and behavior offering an alternative, and possibly complementary, explanation for why these freshwater bottom dwellers emit electric chirps. By uncovering correlations among chirping, electric field geometry, and detectability in enriched environments, we present evidence for a previously unexplored role of chirps as specialized self-directed signals, enhancing conspecific electrolocation during social encounters.

## Introduction

Homeoactive sensing (i.e. self-directed active signal production) differs from alloactive sensing in that energy changes are actively produced in order to affect object detection in the environment, as opposed to a simple reconfiguration of sensor parameters such as orientation or distance from the signal source (*Zweifel and Hartmann, 2020*). Homeoactive sensing is often used as a strategy to enhance spatial navigation in heterogeneous and unpredictable environments: by emitting high-pitch calls (8–200 kHz; *Fenton et al., 1998*), bats are able to detect features of stationary or moving objects while flying at high speeds and in complete darkness (*Jones and Holderied, 2007*). Similarly, odontocete cetaceans are capable of echolocating objects by emitting very short (a few ms long) clicks covering a broad frequency spectrum (5 Hz-100 kHz). A few other mammals (shrews and tenrecs), but

also bird species (oilbirds and swiftlets) and even blind humans (*Griffin, 1944*; *Thaler et al., 2011*), are known to be capable of echolocating target objects through the emission of brief high-frequency echolocation calls (*Konishi and Knudsen, 1979*; *Brinkløv et al., 2013*; *Gould, 1965*). In general, all biosonar systems are good examples of homeoactive sensing strategies and represent a remarkable adaptation to low-light or turbid water environments in which other sensory modalities would offer only a fraction of the information required to optimize fast spatial navigation during foraging or social interactions.

But navigating in the dark is not only about detecting sounds. The electrosensory system of weakly electric fish is known to offer analogous advantages yet at the same time presenting fundamental differences. In the following study, we focus on the brown ghost – one of the most studied species – and extend this analogy to the use of frequency modulations as useful signals in the context of conspecific electrolocation. Brown ghosts belong to the gymnotiform group (as does the electric eel) and occupy benthic habitats in the Amazon river basin (*Crampton, 2019*). They generate quasi-sinusoidal electric fields at relatively high frequencies (600–1000 Hz; *Zakon et al., 2002*) by means of modified motor neurons projecting mainly to the caudal region of the fish body where they form a so-called electric organ. For this reason, these electric fields are also referred to as electric organ discharges (EODs). EODs – and perturbations thereof – are detected through specialized electroreceptors distributed throughout the fish body and mainly concentrated around the head region. Because EODs are actively produced signals, this process is also called *active electrolocation* (as opposed to passive electrolocation, i.e. the sensing of weak bioelectric signals such as those emanating from muscle activity in prey or other organisms; *Rose, 2004*). Compared to echolocation, passive electrolocation is possible only at shorter ranges (0–5 cm; *Lissmann and Machin, 1958*; *von der Emde, 1999*; *Knudsen, 1974*) and, although it is so fast to be considered almost instantaneous, it is more sensitive to signal degradation (*Benda, 2020*; *Nelson and MacIver, 2006*).

Active electrolocation allows to extend the *active space* of electrolocation (*Hopkins and Westby, 1986*; *Brenowitz, 1982*) slightly beyond such short range: since EODs spread roughly equally in all directions, all affected surrounding objects (preys, substrate, vegetation) are simultaneously detected (up to 10–12 cm away from the skin; *von der Emde, 1999*; *Snyder et al., 2007*). During social interactions, the range of active electrolocation can be further extended: when two fish approach each other, the alternating constructive and destructive interaction of their sinewave EODs results in an amplitude- and phase-modulated signal (also referred to as *beat*) detectable by each of them within a 30–40 cm radius (see appendix 1: *Detecting beats at a distance*; *Knudsen, 1975*; *Fotowat et al., 2013*; *Henninger et al., 2018*; *Benda, 2020*). The frequency of such beat modulations is equal to the frequency difference (DF) of the two sinewave signals. Because individual brown ghosts have each a different EOD frequency, detecting the beat can be used not only to sense an approaching conspecific but potentially also to determine its identity and sex: at 27 °C female EOD frequencies range between 600 and 800 Hz while male frequencies are usually between 800 and 1000 Hz. It follows that low DFs may be indicative of same sex encounters, whereas high DFs could signal opposite sex interactions (*Bastian et al., 2001*).

In summary, weakly electric fish are capable of passively detecting bioelectric sources in close proximity to their body (0–5 cm), but also to actively sense inanimate objects as instantaneous EOD perturbations (up to 12 cm; *Chen et al., 2005*; *Snyder et al., 2007*; *Fotowat et al., 2013*). In addition, by detecting beats (i.e. the EOD amplitude modulations induced by other EOD emitting sources), they can instantaneously sense the presence of other conspecifics at distances of up to 30–50 cm (*Fotowat et al., 2013*; *Henninger et al., 2018*; Silva A., personal communication).

Importantly, the strength of electric fields decays with the cubed distance (see appendix 1) and EODs are subject to anisotropic degradation even at very short ranges. This is mainly due to the presence of noisy interferences encountered during locomotion through electrically heterogeneous underwater materials (*Gómez-Sena et al., 2014*; *Pedraja et al., 2016*; *Milam et al., 2019*; *Yu et al., 2019*).

The picture is further complicated by the presence of another critical source of beat interference: communication signals. These consist of brief EOD frequency modulations – or *chirps* – which can be broadly classified in at least 4 different types, based on their duration and the extent of the chirp frequency modulation (FM, *Engler et al., 2000*; *Zupanc et al., 2006*). Different types of chirps are thought to carry different semantic content based on their occurrence during either affiliative

**Table 1.** Chirp categories .

| Chirp type | FM (Hz) | Duration (ms) | References |
|---|---|---|---|
| type 1 | 200–400 | 25 | *Engler et al., 2000*; *Bastian et al., 2001*; *Zakon et al., 2002* |
| type 2 | 50–100 | 15–20 | *Engler et al., 2000*; *Bastian et al., 2001*; *Hupé et al., 2008* |
| type 3, 4 | 200–300 | 75–200 | *Hagedorn and Heiligenberg, 1985*; *Engler et al., 2000*; *Engler and Zupanc, 2001* |
| rises | 10–20 | 50–100 | *Engler and Zupanc, 2001*; *Tallarovic and Zakon, 2002* |

or agonistic encounters (this notion will be referred to as *communication-hypothesis*; *Larimer and MacDonald, 1968*; *Bullock, 1969*; *Hopkins, 1974*; *Hagedorn and Heiligenberg, 1985*; *Zupanc and Maler, 1993*; *Engler et al., 2000*; *Engler and Zupanc, 2001*; *Bastian et al., 2001*).

Since both chirps and positional parameters (such as size, orientation, or motion) can only be detected as perturbations of the beat (*Petzold et al., 2016*; *Yu et al., 2012*; *Fotowat et al., 2013*), and via the same electroreceptors, the inputs relaying both types of information are likely to overlap and interact in highly variable ways (as noted also by *Field et al., 2019*). Moreover, as the majority of chirps are produced within a short range (<50 cm; *Zupanc et al., 2006*; *Hupé et al., 2008*; *Henninger et al., 2018*; see appendix 1) this interference is likely to occur consistently during social interactions.

Under the *communication-hypothesis*, the assumption that chirps and beats are conveying different types of information (i.e. semantic value as opposed to position and related geometrical parameters) is therefore leaving this issue unresolved. (*Table 1*).

In this study, we propose a solution to this problem by providing evidence for a previously unexplored function of chirping in weakly electric fish. We have gathered a large dataset of circa 67.000 chirps, obtained from staged social pairings, playback experiments and behavioral assessments of fish locomotory activity (see *Table 2*). We first show that the relative number of different chirp types in a given recording does not significantly correlate with any particular behavioral or social context. It is instead correlated to the biophysical properties of underwater electric fields and to the spatial arrangement of interacting EODs. By using high-speed infrared video recordings synchronized with multi-channel voltage data acquisition, we then analyzed chirps considering both their effects on beat processing and their spatial attributes, that is the spatial location of the interacting fish. Analyses of cross-correlation of chirp time-series, chirp-type transitions and of behavioral responses to playback chirps were used to assess the presence of meaningful correspondences. The occurrence of chirps of different types was then analyzed in relation with the most common behavioral displays used by brown ghosts during social interactions. Finally, by recording fish in different conditions of electrical 'isibility', we provide evidence supporting a previously neglected role of chirps: homeoactive sensing.

## Results

### Experience is the main factor affecting chirp variability

During social interactions, communication signals are often related to immediate aspects of behavioral physiology (e.g. internal states, *Owren et al., 2011*). As a consequence, correlations between

**Table 2.** Chirp dataset (total = 67,522 chirps).

| Experiment | Figure | N fish | Total | Type1 | Type2 | Type3 | Rises |
|---|---|---|---|---|---|---|---|
| fish pair interactions | 1, 2, 5 | 130 | 30,486 | 4842 | 23,169 | 2395 | 80 |
| playback chirps | 3 | 16 | 15,720 | 2428 | 12,880 | 331 | 81 |
| playback freq ramps | 4 | 14 | 3966 | 779 | 3012 | 89 | 86 |
| freely swimming pairs | 6 | 24 | 4672 | 252 | 4164 | 77 | 179 |
| novel environment | 8 | 30 | 7893 | 1059 | 5172 | 239 | 1423 |
| cluttered environment | 9 | 12 | 1864 | 529 | 1225 | 80 | 32 |
| cluttered playback | 9 | 8 | 2921 | 405 | 2258 | 241 | 17 |

signals and behavioral contexts are often indicative of signal meaning (*Flack and Waal, 2007*). Thus, if behavioral meaning can be attributed to different types of chirps, as posed by the *communication-hypothesis* (e.g. *Hagedorn and Heiligenberg, 1985*; *Larimer and MacDonald, 1968*; *Rose, 2004*), one should be able to identify clear correlations between behavioral contexts characterizing different internal states and the relative amounts of different types of chirp.

To assess the presence of correlations between different behavioral contexts and the emission of different types of chirps, we recorded fish pairs in different settings based on (1) the experience with the recording environment (*residents, intruders* and *equal,* to define cases in which both fish of the pair were naïve to the environment), (2) the behavioral context including: dominance status (*dominant* and *subordinate*), simulated breeding conditions (*breeding* or *no breeding*), social experience (*experienced, novel*), and the direct physical accessibility of a conspecific (*divided* or *free*) for a total of N=130 fish pairs (see Materials and methods for details).

EOD recordings took place in chambered aquariums using a multi-channel acquisition setup (*Figure 1A*). Chirp detection was carried out using a semi-automatic working pipeline (*Figure 1B*, see Materials and methods). Chirps have been previously categorized in different types based on the extent and the duration of their frequency modulation (FM in Hz and duration in ms; *Engler et al., 2000*). Initially, we searched for clusters in our chirp dataset using the k-means method and chirp FM and duration as input parameters (*Figure 1C*). Through this approach, 2 clusters were identified (clustering was validated using the *silhouette* method; *Rousseeuw, 1987*), suggesting a roughly bimodal distribution of chirps, according to these basic features (a third cluster could be defined by the separation between type 1 and type 2 chirps, which in our sample was not so obvious). Nonetheless, for consistency with the existing literature, we categorized chirps using rough cut-off values based on the distribution density of an early subset of chirps (N=11,342 chirps, see Materials and methods for details). These values roughly matched the broad parameter ranges previously reported for each type of chirp (*Engler et al., 2000*; *Zupanc et al., 2006*; *Zakon et al., 2002*; *Bastian et al., 2001*). Based on these cut-off values (duration = 50ms, FM = 105 Hz), chirps were categorized in short (with small - type 2 - or large - type 1 - FM) and long (with small - rises - or large - type 3 - FM; *Figure 1C*). Slower events, such as gradual frequency changes (or *gradual frequency rises*; *Engler et al., 2000*), were more sporadic and were not considered in this analysis.

Overall, the majority of chirps were produced by male subjects, at rates that seemed affected by environmental experience (*resident, intruder* or *equal*; *Appendix 1—figure 2A, C*), social status (*dominant* or *subordinate*; *Appendix 1—figure 2B*) and social experience (*novel* or *experienced*; *Appendix 1—figure 2D*). Interestingly, this sex difference was reversed when fish were allowed to interact freely, as opposed to being separated by a plastic mesh divider (*Appendix 1—figure 2E, F*). Sex differences were found also in the types of chirp produced, mainly in the case of male-to-female exchanges (MF; *Appendix 1—figure 2C, D*), confirming previous accounts of marked sex dimorphisms in both the number and the type of chirps produced by brown ghosts (*Dye, 1987*; *Zupanc and Maler, 1993*; *Dulka and Maler, 1994*).

To better define the main factors underlying chirp variability (i.e. types, frequency of beat and carrier signals, behavioral context, etc.), we used a 3D factor analysis for mixed categorical and numerical data (FAMD) and evaluated for each recorded chirp several parameters relative to the interacting fish (both *sender* and *receiver*) and their respective EODs (see Materials and methods for details; *Figure 2*). The model obtained was adjusted in order to include the most significant variables (those explaining above 20% of the total variance). The lowest number of variables required to explain the highest percentage of variance (N=3: *tank experience*, *DF* and *context*) was obtained by further simplifying the model through subsequent iterations (percent of variance explained 52.3%).

Overall, this analysis indicated that environmental and social experience, together with beat frequency (DF) are the most important factors explaining chirp variability (*Figure 2B*). It also indicates that chirp parameters such as duration and FM do not seem to be associated with any particular context in a meaningful way, other than being affected by beat frequency. The plot of individual chirps (*Figure 2C*) shows the presence of clustering around different categorical variables and it reveals that experience levels or swimming conditions are important factors affecting chirp distribution (note for instance the large central 'breeding' cluster in which fish are *divided* and the smaller ones in which fish are *free*). Sender or receiver identity does not individuate any clear clustering relative to either sex (see the overlap of male_s/male_r and female_s/female_r) or social status (dominant/subordinate).

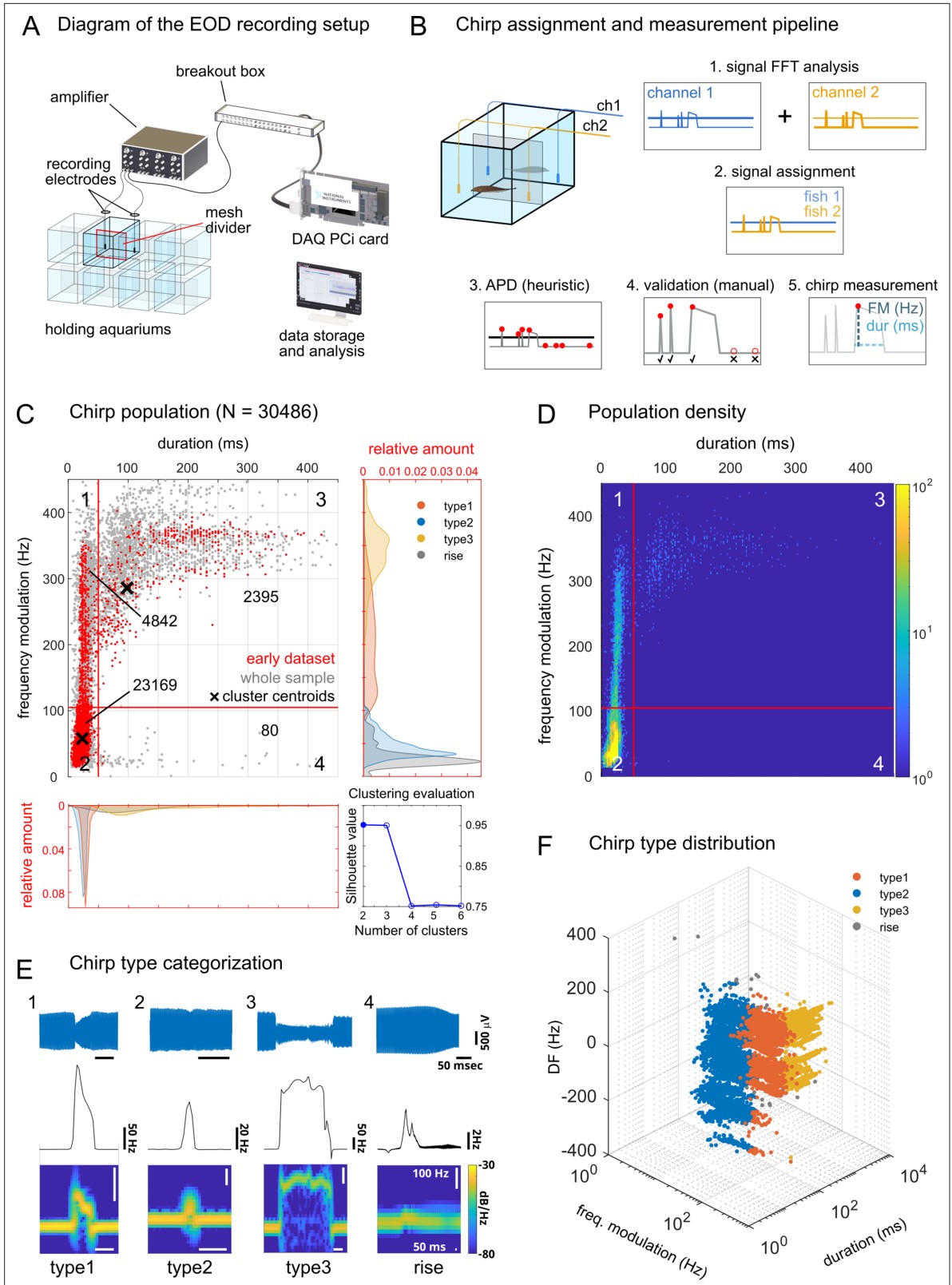

**Figure 1.** Chirp type categorization based on FM parameters. (**A**) Schematic diagram of the recording setup: holding aquariums (N = 8) were divided by a plastic mesh divider to prevent physical contact between the interacting fish (N = 2 per aquarium) while allowing electric interactions. Brown ghost EODs were detected by electrodes placed in each tank compartment (2 input channels per tank), amplified, digitized (20 kHz sampling rate) and recorded using custom-written MATLAB scripts. (**B**) Electric signals and chirps were analyzed using their fast-fourier transforms (FFT, 1) and assigned

*Figure 1 continued*

to different fish based on the signal intensity (2). Frequency modulations were detected using a MATLAB-based heuristic method for automatic peak detection (APD, 3). False-positives, false-negatives or wrongly assigned chirps were revised manually (4) and chirp FM and duration were then measured (5). (**C**) Chirp distribution by frequency modulation (Hz) and duration (ms). K-means chirp clustering indicated an optimal value of 2–3 for clustering chirps based on these two parameters (cluster centroids are indicated with black crosses; silhouette values are shown in the bottom right panel). The red lines indicate the cut-off values used to classify the different chirp types (type 1 = 1, type 2 = 2, type 3 = 3, rises = 4): duration 50ms and modulation 105 Hz. These values are based on a dataset acquired at the beginning of the study (red, in the scatter plot; N chirps = 11342; N chirping fish = 16, N fish pairs = 8) and on the distribution density (**D**) of the whole chirp population (gray, in the scatter plot; N chirps = 30486; N chirping fish = 130, N fish pairs = 78). (**E**) Representative examples of the 4 different chirp categories (voltage data on the first row, instantaneous frequency on the second and spectrogram on the third). (**F**) Scatter plot showing the distribution of different chirp types by DF (frequency difference between sender and receiver fish): note the gradual change in chirp type composition (color coded) at different DF values (especially visible for type 1–3). Due to the sex difference in the brown ghost EOD frequency, negative DF values correspond mainly to females, positive values to males.

Chirps labeled based on tank experience (i.e. *resident* vs *intruder*) are instead clearly separated. Social experience may also be relevant although it appears to affect chirp clustering only slightly (note the contiguity of the *novel* and *experienced* clusters). Nonetheless, most of the clustering could be simply explained by the DF (see inset in *Figure 2C*), as suggested by the lack of association of chirp types to the other social variables considered (*Appendix 1—figure 3*).

## Brown ghosts display invariant chirp responses to chirping playback EOD mimics

A complementary approach to uncover correspondences between communication signals and behavioral states relies on the use of playback experiments. To determine whether brown ghosts respond to playback chirps in a chirp-type specific manner, we adopted a variation of the classic 'chirp chamber' approach (e.g. *Dye, 1987*; *Bastian et al., 2001*; *Dunlap and Larkins-Ford, 2003*) and recorded brown ghosts' electrical and locomotor responses to playback EODs in freely moving conditions (N=16 fish, 8 females, 8 males; *Figure 3A*). In these experiments, the matching of emitted signals to playbacks as well as the locomotor activity (video recorded) displayed during close-range interactions with EOD sources (i.e. decisions to approach or avoid playback signals), can be used as a possible measure of signal value (*Fugère and Krahe, 2010*; *King and McGregor, 2016*).

Fish were allowed to swim in a 30x80 cm aquarium at whose opposite ends two 13x30 cm playback compartments were created using plastic mesh barriers. Playback electrodes were positioned in either compartment (randomly chosen) and parallel to the longest tank axis, ca. 10 cm apart. Electric stimuli were EOD mimics at different frequency steps (DFs) of both positive and negative sign, relative to the fish's own EOD frequency (values in Hz:±240,±160,±80,±40,±20,±10,±5, 0). The EOD playback intensity was set at 1 mV/cm at ca. 5 cm distance (which is in the range of natural field strengths of this species). Playback sessions were organized in four different *modes*, based on the pattern of frequency modulations contained: mode 0=no chirps, mode 1=type 3 chirps, mode 2=type 2 chirps, mode 3=rises. Chirp parameters were set to represent rises and the two main chirp categories defined by the k-means clustering without the constraints of our cut-off values (*short* – type 1 and type 2 – and *long* – type 3 – chirps; *Figure 1C and D*). Each playback trial lasted ca. 60 s with 180 s inter-trial intervals (see Materials and methods for details). An infra-red video camera synched with the EOD recording was used to score the average fish position during trials (see Materials and methods).

Fish responses to plain sinewave EODs confirmed previously reported findings: fish respond with large chirps (type 1 and type 3) to high DFs and with small chirps (type 2) to low DF signals (*Appendix 1—figure 4*), with males producing more chirps than females (*Figure 3B*; *Bastian et al., 2001*; *Engler and Zupanc, 2001*). Notably, a transient decrease in chirping was observed in the immediate temporal surround of playback chirps (modes 1–3, *Appendix 1—figure 5*). This effect was similar for all chirp types and seemed to be limited to a short time around chirp occurrence. Accordingly, a more pronounced decrease in chirping was observed during 3 Hz trains of type 2 chirps, possibly due to the higher frequency at which these events were repeated (*Appendix 1—figure 5*). Apart from this, fish responses to chirping EODs were hardly distinguishable from non-chirping EOD mimics in terms of number of chirps produced (*Figure 3C and D*; but see also *Triefenbach and Zakon, 2008*).

Analysis of the time course of chirping further revealed two interesting details: first, while type 1 and type 3 chirps seem to be randomly scattered throughout the whole trial (60 s), the number of

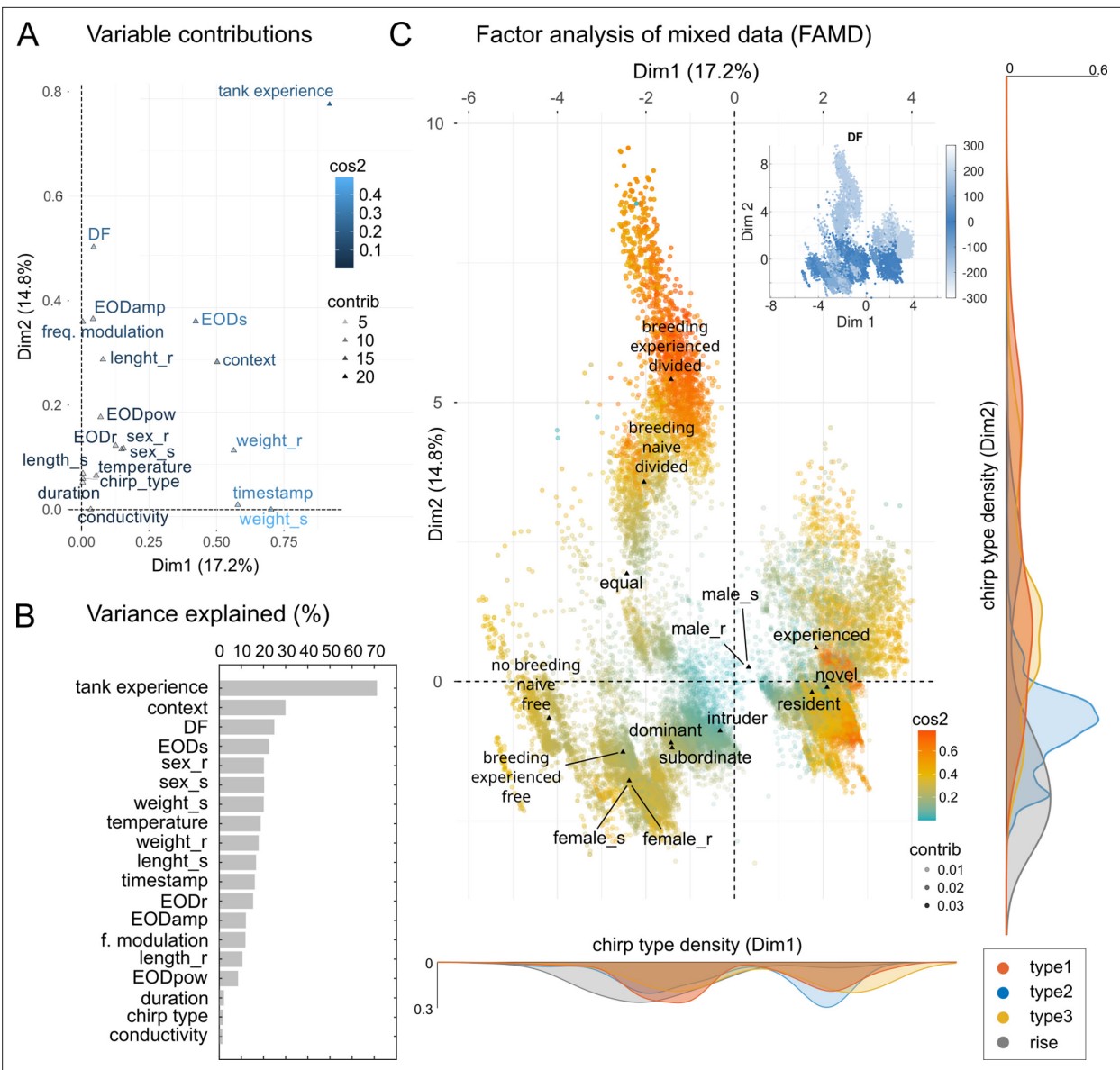

**Figure 2.** Factor analysis of mixed data (FAMD) – social contexts and tank experience. (**A**) Scatter plot showing the contribution of all chirp-related variables to the overall variance of the whole chirp dataset: among these, EOD parameters such as amplitude (EODamp), frequency (EODs or EODr, based on sender or receiver identity), spectral power density (pow) were considered together with variables related to chirps, such as frequency modulation (freq. modulation), duration, sex of sender or receiver fish (sex_s, sex_r), the time of occurrence within a 1 hour trial (timestamp), the type (1–4), the DF. Variables related to the fish experience with either the tank environment (tank experience: resident = 1-week tank experience, intruder = new to the tank, equal = both new) and experience with the paired conspecific (context). This latter category refers to the reciprocal experience of each fish pair (novel or experienced), their hierarchical status (dominant or subordinate), the type of interaction (divided = behind a plastic mesh barrier, free = freely swimming) and the simulated breeding season (based on water conductivity levels: high conductivity = ca. 400 µS, no breeding; low conductivity = ca. 100 µS, breeding) at which the interaction takes place (see Materials and methods for details). Triangles indicate the coordinates of the variable centroids, their contribution ('contrib') is coded by color intensity, whereas the quality of their representation on the transformed coordinates is coded by color hue ('cos2'). (**B**) Estimates of the total variance explained indicate that tank experience, together with DF and context, are the most important factors explaining chirp variance. (**C**) Representation of chirps in the transformed coordinates. The clustering is based on qualitative coordinates (tank experience, context and chirp type). Cluster distance represents the correlation among variables. The marginal plots show the kernel distribution of the chirp population color-coded according to chirp type (legend on the bottom right). Labeling chirps by DF shows how chirps can be meaningfully clustered based on this parameter (inset, top right).

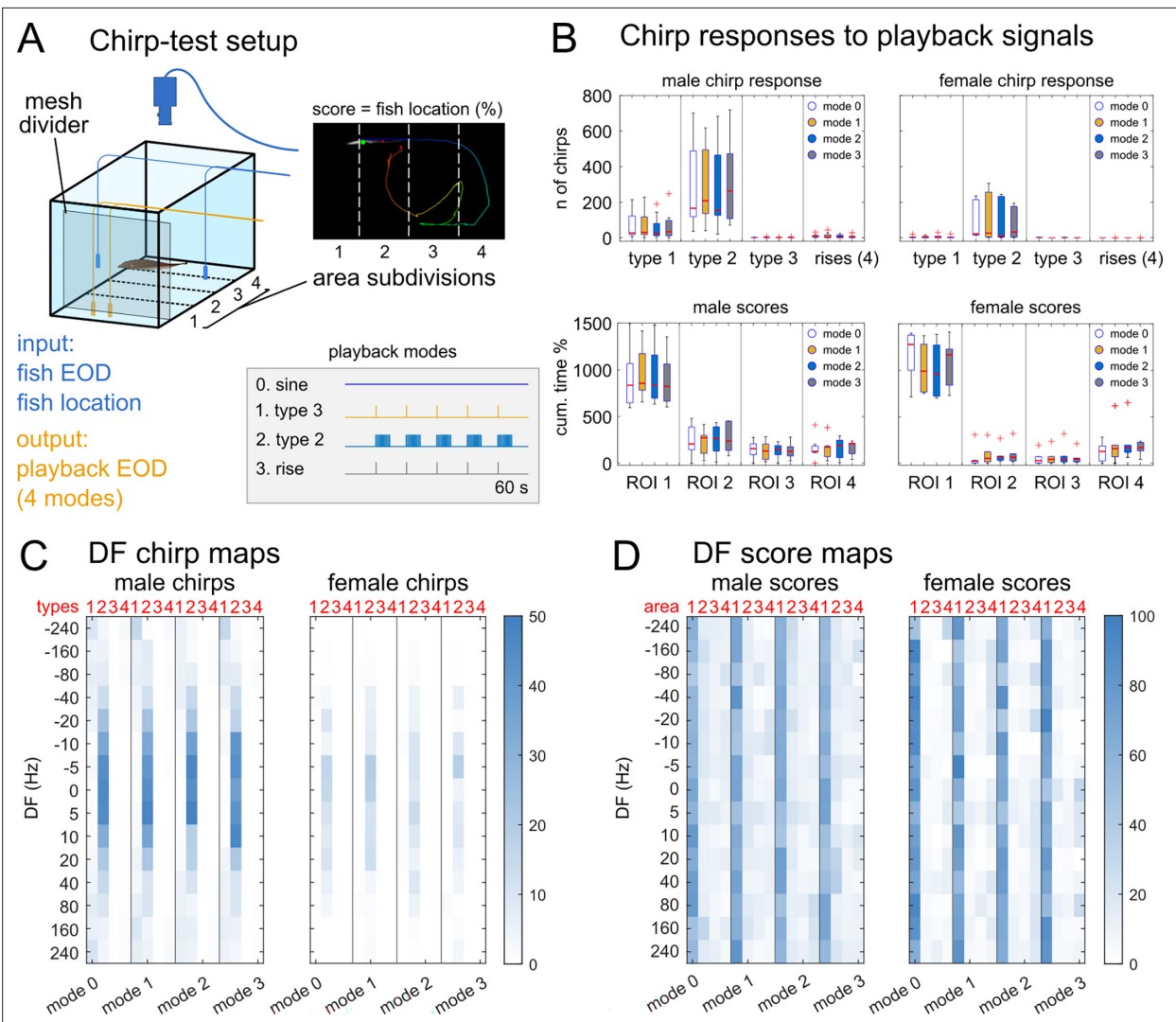

**Figure 3.** Invariant chirping responses to playback chirps in freely swimming fish. (**A**) Schematic diagram of the setup used for the playback experiments: both the fish EOD and the fish swimming behavior are recorded during 60 s long playback trials. During each trial the fish locomotion is scored based on the % coverage of tank space (60 x 30 cm) in 4 regions of interest (ROI) at increasing distance from the playback electrodes (1 = close, 2-3 = intermediate, 4 = far). Playback trials are organized in 4 different modes (0–3), each including 15 DF levels, all shuffled and randomized. (**B**) The box plots on the top row show the total number of chirps produced by either male or female brown ghosts produced in response to plain EOD mimics (mode 0), sinewaves containing type 3 chirps (red, mode 1), sine waves containing trains of type 2 chirps (blue, mode 2) or rises (grey, mode 3). Chirp counts relative to each individual fish are summed across different DFs. The boxplots on the bottom row show the trial scores relative to the same subjects (i.e. the cumulative sum of the percentage of time spent in each ROI) and summed across different DFs and modes. (**C**) Heatmaps showing the DF-dependent but mode-invariant distribution of mean chirp types produced by male and female subjects in response to different playback regimes. (**D**) Score heatmaps showing that fish of both sexes approach playback sources (i.e. higher scores in ROI 1) with equal probability regardless of playback types or DFs. Female fish may be more stationary in proximity of the electrodes, resulting in slightly higher ROI-1 scores.

type 2 chirps seems to increase over time (*Appendix 1—figure 6*; as reported by *Hupé et al., 2008*). This trend was independent of the type of playback and is therefore intrinsically related to the use of this particular type of chirp. All chirp types are subject to a brief decrease only in concurrence with 2 Hz trains of type 2-chirp playback (*Appendix 1—figure 6*). Second, the vast majority of rises was produced right at the end of the playback signal, regardless of whether or not playback chirps were occurring (*Appendix 1—figure 6*).

Overall, these results show that brown ghosts do not respond to playback chirps in a chirp-type specific manner. They are coherent with previous reports showing an increase of type 2 chirps over time during paired interactions (*Hupé et al., 2008*) and the presence of a short delay between consecutive

chirps produced by different subjects (*Hupé et al., 2008*; *Henninger et al., 2018*). The consistent occurrence of rises at the end of playback signals has also been reported previously (*Kolodziejski et al., 2007*). Eventually, the strong symmetrical dependency of chirp production on positive and negative beat frequencies (DFs; *Figure 3C*, *Appendix 1—figure 4*; *Bastian et al., 2001*; *Engler and Zupanc, 2001*; *Triefenbach and Zakon, 2008*; *Kolodziejski et al., 2007*) and the lack of specific matching in the response of brown ghosts to playback chirps confirms that, in this species, chirping is primarily conditioned by beat parameters (*Walz et al., 2013*). Finally, these findings indicate that, irrespective of the meaning or valence chirps may have, the production of chirps and the detection of non self-generated chirps may be two independent processes.

Eventually, if chirping depends merely on beat parameters (such as beat frequency, i.e. DF, or contrast) one would expect to observe invariant responses to stimuli of different duration as well as across repeated trials, because the emission pattern of chirps of different types would adjust to frequency changes in similar ways across trials.

To evaluate this possibility, we stimulated a group of brown ghosts with playback signals with frequency ramps (DF +/-300 Hz, in both directions). Stimuli were provided with different durations (20, 60, 180 s), each in 3–4 repeats. Individual fish were caged in a mesh tube and playback stimuli were calibrated as previously done.

Perhaps not surprisingly, we found that chirps are produced at higher rates when playback frequency gets closer to the fish's own EOD frequency (i.e. reaching a low DF value), similarly to what happens when static frequency playbacks are used. Yet, the choice of chirp type seemed to be highly variable across individuals (*Figure 4D–F*). Despite this heterogeneity, each individual fish displayed rather similar chirping patterns, both across trials of different length as well as in repeats of the same trial (*Figure 4D–F*). In addition, regardless of the type of chirp produced in a given trial, the overall number of chirps was reliably correlated with trial duration with an astonishingly high linear correlation coefficient ($R^2$=0.987, *Appendix 1—figure 7*). The relative number of the different types of chirps also varied depending on playback duration, possibly as an effect of ramping speed (pie charts in *Figure 4G–I*). As previously shown, chirp type choice on average appears to reflect changes in the DF sign: larger chirps are scarcer around 0 Hz, while smaller ones are more frequent around the smaller DF values (*Figure 4G–I*). In some cases, this implied a quick DF-dependent transition from large to small chirps (see fish IDs S6, G5, and P3 for instance, *Figure 4D–F*). Nonetheless, our results show a general trend for all chirp types to be produced more often around lower DF values, as evident by looking at inter-chirp intervals (ICIs, *Appendix 1—figure 8*). The reliability by which the chirping response adapts to both the rate and direction of beat frequency is variable across individuals but rather stable across trials (relative to a given subject), further suggesting that chirp type variations may not reflect changes in internal states or in the animal motivation to specific behavioral displays (presumably subject to less abrupt variations and stereotypical patterning based on DF). Instead, the choice of different chirp types may reflect the presence of different sensory demands posed by different beat frequencies to the processing of the beat itself (*Barayeu et al., 2023*). Taken together, these results indicate that chirp patterning is mainly affected by beat frequency (i.e. DF).

## Chirping is mainly a self-referenced behavior

As shown in other communication systems (bats, *Bohn et al., 2008*; cetaceans, *Dunlop, 2017*; primates, *Slocombe and Zuberbühler, 2005a*; *Slocombe and Zuberbühler, 2005b*; *Notman and Rendall, 2005*), valence and behavioral meaning could reside in conserved motifs that can serve as unequivocal signifiers for the specific context in which they are produced (*Berwick et al., 2011*; *Nieder and Mooney, 2020*). Similarly, in brown ghosts, while single chirps may neither be behaviorally salient nor informative, meaning could be attributed to chirp patterns or chirp transitions produced by the senders of the signals, the receivers or both. In agreement with this idea, in weakly electric fish, specific chirp sequences have been proposed to be relevant to specific behavioral contexts (*Henninger et al., 2018*; *Dye, 1987*) and even emotional states (*Triefenbach and Zakon, 2008*; *Smith, 2013*; *Silva et al., 2013*).

To validate this idea and assess whether specific chirp patterns could be considered signatures for different types of behavioral interactions, we evaluated all possible chirp sequences in the chirp time-series recorded in our first set of experiments (*Figure 1*). This was done first independently

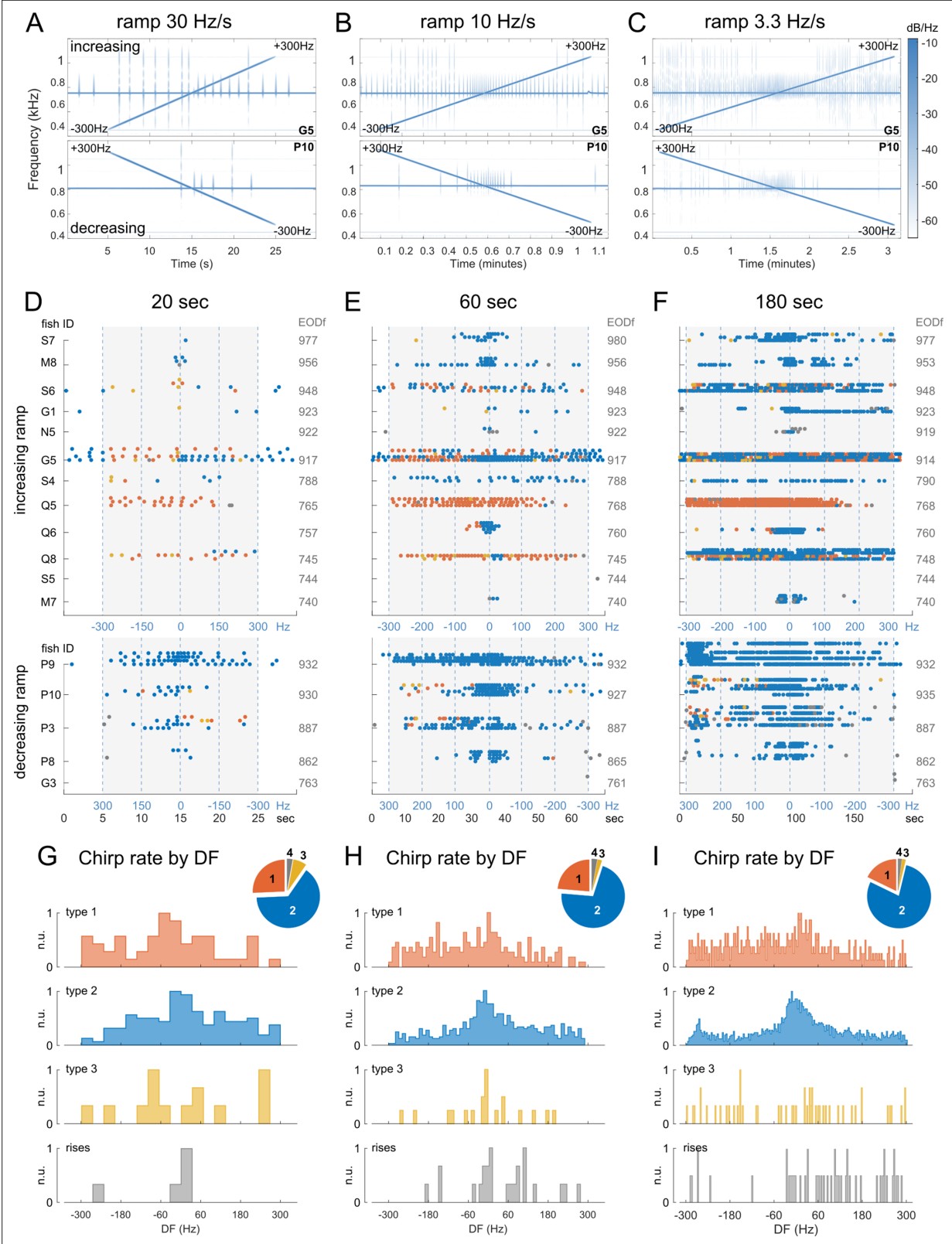

**Figure 4.** Responses to EOD frequency ramps. (**A–C**) Examples of spectrograms from playback trials showing the responses to frequency ramps (increasing from –300 Hz to 300 Hz or decreasing within the same range) of different durations: 20, 60, and 180 s, respectively. (**D–F**) Chirp raster plots relative to the three trial types. Chirp responses are grouped based on fish identity (unspaced rows represent responses by the same fish). Playback time is represented by the gray areas. Some fish produced chirps even in absence of a playback EOD. (**G–I**) Histograms showing chirp type distributions by

*Figure 4 continued on next page*

*Figure 4 continued*

DF. Trials in which decreasing ramps were used were adjusted by flipping the time array to match the DF values. The pie plots show the relative amounts of the four types of chirp.

(considering each subject separately; *Appendix 1—figure 3B, D*) and then interactively (considering sender-receiver transitions; *Figure 5*).

Since this analysis led to very similar results for different chirp sequence lengths (2, 4, 6, 8 chirps), for simplicity we focus here on the simplest case of the pairwise transitions (i.e. sequence length = 2).

Considering all recordings independently, type 2 chirps are overall the most frequently produced chirp type, both in females as well as males (*Appendix 1—figure 3A, C*). As a direct consequence, the most common chirp type transition found is 'type 2'-'type 2', except for the case in which males are chirping to females (labeled as MF; *Appendix 1—figure 3B*). In this case, repeats of type 1 chirps become more prominent (especially in experienced pairs), possibly due to the higher amount of type 1 chirps produced by males at high frequency beats (i.e. high DF; *Figure 1F*, *Appendix 1—figure 3C*) and to the concurrent decrease in the number of type 2 chirps produced by experienced fish (see the effect of *chirp timestamp* in *Figure 2A* and experience in *Appendix 1—figure 2D*). In any other case, neither chirp transitions nor chirp type ratios were differently distributed. Rather, the proportion of different chirp types and the transition frequencies from each type to the others were remarkably similar, regardless of the context and the sex of the interacting fish (*Appendix 1—figure 3*). Overall, these results show that, at an individual level, the most common chirp sequence is a repeat of type 2 chirps in both males and females, with type composition being only slightly more heterogeneous for male senders, when paired with female receivers. This further supports the idea that chirp type choice depends mainly on beat frequency (DF). In other words, at least when chirp sequences are considered individually, chirp-type patterning is highly conservative and not connected to any of the behavioral contexts considered, except the one identified by low vs high DFs.

Yet, it is possible that meaningful chirp transitions (i.e. correlated to a specific behavioral context) are to be identified only when considering time series interactively (*Chow et al., 2015*; *Fröhlich et al., 2017*). Recordings from brown ghosts, conducted both in the field and in captivity, suggested the presence of correlations between male chirp rate (type 2 and type 3) and female larger chirps (type 3; *Henninger et al., 2018*). Meaningful correlations have been also suggested to consist more broadly in temporal correlations between time-series, with chirps produced with a preferred latency by two interacting subjects. However, different authors reported different latency values (*Zupanc et al., 2006*, latency 500–1000ms; *Hupé et al., 2008*, latency 200–600ms; *Henninger et al., 2018*, latency 165ms). These discrepancies could be due to either species differences or differences in the recording settings.

In our recordings, although the temporal correlation of chirp series was highly variable among fish pairs (*Figure 5*), it often on the number of chirps produced (as shown also by estimates of the maximum chirp rates in *Zupanc et al., 2006*, chirp rate 0.4 /s; *Hupé et al., 2008*, chirp rate 0.53 /s; *Henninger et al., 2018*, chirp rate 3 /s) raising the question of whether the previously reported temporal correlations could be just a direct consequence of higher chirp rates occurring when fish get closer to each other (*Zupanc et al., 2006*). We assessed chirp transitions considering both senders and receivers interactively for different sex pairings (M-M, M-F, F-M, F-F; *Figure 5A–D*). Transition frequencies were mapped onto four quadrants defined by fish identity (1 or 2) and temporal order: 1–1, 1–2, 2–1 and 2–2.

What stood out most consistently from this analysis was the much lower (or sometimes completely absent) extent of sender-receiver chirp transitions, as shown by the higher transition probabilities along the 1–1 and 2–2 quadrants of the probability maps. According to this, cross-correlation indices are very low or near zero levels in all cases (*Figure 5*). Although slightly different interaction rates were observed in different contexts, in all cases the most frequent chirp transitions were self-referenced (1–1 or 2–2). Although within FF pairs 2–1 and 1–2 interactions seem to be more frequent (*Figure 5C, F and H*), this is probably due to the much lower number of chirps produced by this sex (see boxplots showing the total number of chirps in each group).

Eventually, the absence of conserved temporal correlations could also depend on the type of behavioral interactions and the proximity at which they take place (*Dunlap, 2002*). Social distance in particular could explain the high chirp-rates previously observed during courtship (*Hagedorn and*

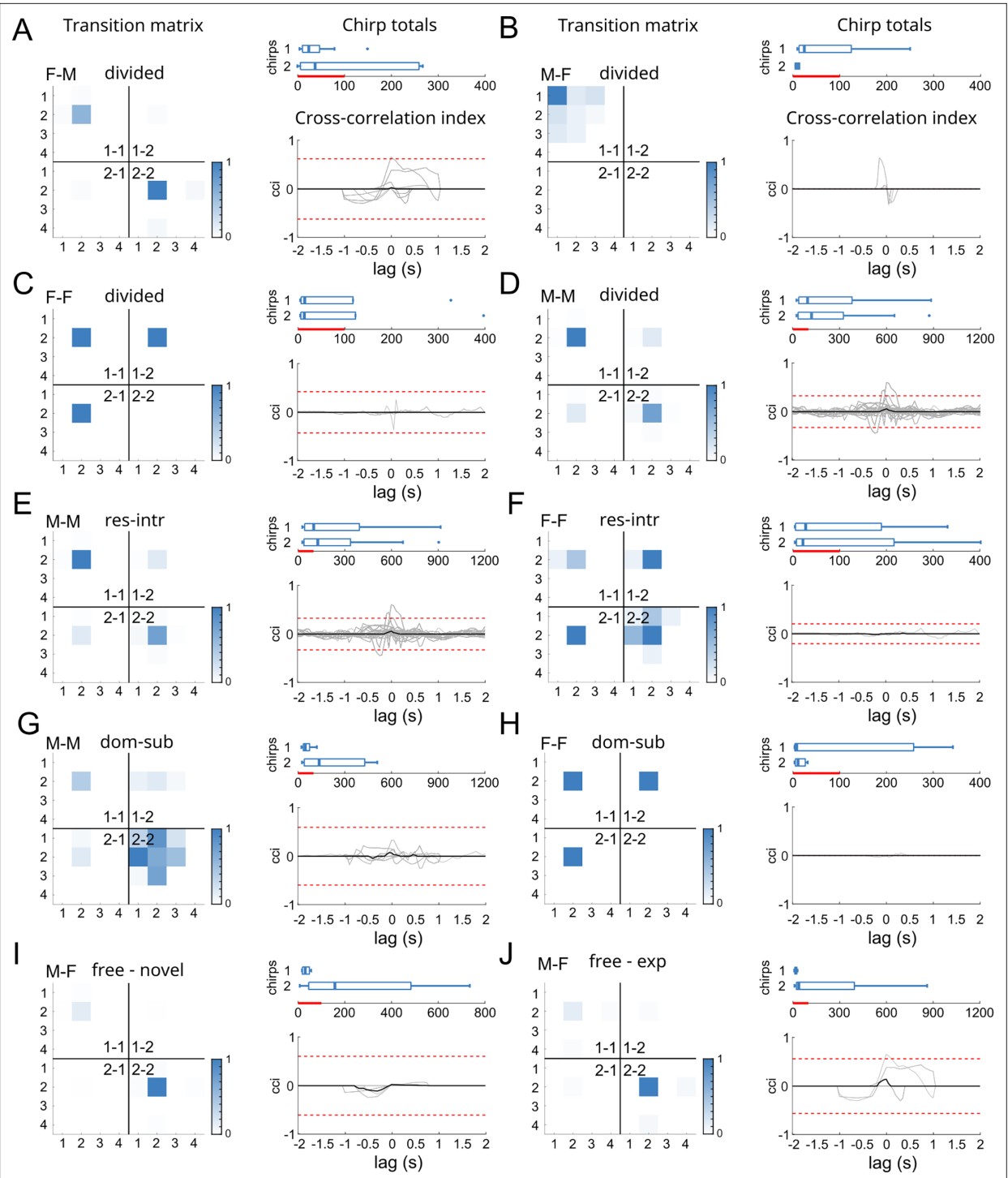

**Figure 5.** Chirp time-series correlations during social interactions. (**A, B**) Chirp transition maps representing the median transition probability (normalized) of all possible chirp type pairs calculated for female and male fish, considered together. For each type of social pairing, the identity of the two fish is indicated by different ID numbers in the different map quadrants: 1–1 = fish 1 to fish 1, 1–2 = fish 1 to fish 2, 2-1 = fish 2 to fish 1, 2-2 = fish 2 to fish 2. In mixed pairs, the order given in the sex tag of each plot follows the same order of the fish ID. The presence of higher chirp-transition frequencies in the 2nd and 4th quadrants of the matrices (labeled with 1–1 and 2–2) indicates a substantial independence between chirp time-series (i.e. lack of temporal correlation). Transitions from female to male fish (**A**) or male-to-female fish (**B**) are considered separately. For each case, chirps were selected based on the sex of the sender fish. The 1–2 numbering refers to fish identity. On the right side of each matrix, chirp totals are displayed in boxplots for sender and receiver fish (outliers are represented as dots lying beyond the boxplot whiskers; red bars = 100) and cross-correlation indices (cci, 50 ms binning) are provided for chirp time series relative to the same fish pairs (red dotted lines = confidence intervals corresponding to

*Figure 5 continued on next page*

*Figure 5 continued*
3 cci standard deviations). (**C, D**) Chirp transitions relative to same sex pairings. A higher level of interaction for F-F pairs (visible in the first and third quadrants in C) is probably due to the extremely low chirp rates in these pairs. Notably, higher chirp rates (as in M-M pairs, D, E) do not result in higher cross-correlation levels. (**E, F**) Chirp transitions for resident-intruder pairs (M–M and F–F). Since most chirps are produced by M-M resident-intruder pairs in divided aquariums, plots in E resemble those in D, as they are relative to overlapping datasets. (**G, H**) Chirp transitions for dominant-subordinate pairs (M–M and F–F). (**I, J**) Chirp transitions for freely swimming (naive, I or experienced, J) opposite-sex pairs. Note the reversed sexual dimorphism in chirp rates, in both cases.

Heiligenberg, 1985; Henninger et al., 2018) and short-range agonistic interactions between fish confined in a relatively small aquarium (Hupé et al., 2008) as opposed to the more sporadic chirping observed during encounters over wider spatial ranges (Henninger et al., 2018). Alternatively, the higher chirp temporal cross-correlation previously observed by other authors could be due to neuro-endocrine effects of courtship or other mating-related factors.

To test whether cross-correlation would be significantly affected during courtship, we simulated the onset of the reproductive season by gradually lowering water conductivity over the course of a 40-day period (from 400 µS to 100 µS, Kirschbaum, 1979). As a result of this treatment, at least 75% of the females used in this experiment (N=8) showed visible signs of egg maturation. Mixed-sex pairs (N=8 pairs) held in such conditions were then allowed to interact freely, in absence of any tank divider. As a control group we paired males and females in absence of any water conductivity change ('novel' vs 'experienced', *Figure 5I and J*). Although female fish produced a surprisingly higher number of chirps in these conditions, only minimal signs of cross-correlation could be detected, without reaching on average the threshold for statistical significance (*Figure 5J*). Collectively, these results indicate that brown ghosts utilize a very limited variety of chirp type transitions in different behavioral contexts and do not emit specific chirp patterns in any given behavioral context analyzed. More importantly, they show that – at least in the social conditions analyzed here and within small-sized time windows – chirp time series produced by different fish during paired interactions are consistently independent of each other. This further confirms the idea of chirp production and detection as independent processes and that chirp type variability merely depends on beat frequency (DF).

## Chirping correlates with active sensing behaviors

To gain better insights into the behavioral correlates of chirping, we conducted simultaneous video and EOD recordings of freely swimming fish pairs (N=12, mixed sexes; see Materials and methods). The aim of these experiments was to assess the occurrence of consistent chirp-behavior correlations and more in general to evaluate chirp patterning in conjunction with behaviors other than chirp production. Behaving fish were recorded using an infrared camera under IR illumination (920 nm). Fish were allowed to habituate to the testing conditions (15 min) prior to recording of both behavioral and electrical activity (15 min). In this phase, fish were allowed to sense each other electrically through a plastic mesh divider. EODs were recorded in this phase and after removing the mesh barrier. Of the 23 fish chirping during these trials, 12 increased chirping once the direct interaction started, 5 did not alter their chirping rates, 6 decreased them. Chirps were coded again in four types. The ethogram used for this analysis was created according to previously published observations (*Triefenbach and Zakon, 2008*) and included the following behaviors: attacks (i.e. bites to the fish head, *atk*), chasing attempts (*ch*), escape attempts (*esc*), head butts (i.e. direct hits of the snout to the target fish flank, *head*), jaw quivering (often preceding attacks directed to the head or the mouth, *jaw*), backward swimming (*knife*), forward locomotion (*loc*), stationary motion or inactivity (*rest*), backward approaches with tail contact (*tail1*) or involving tail waving (*tail2*), and close body contact (not always aggressive in nature, accompanied by side-to-side swimming, body intertwining and even periods of inactivity, *wrestle*).

Peristimulus time histograms (PSTH) calculated around the annotated chirps (N=4672 chirps, of which 252 were type 1, 4164 were type 2, 77 were type 3 and 179 were rises) showed no behavior reliably occurring immediately after or before chirping (time window = 4 s). Chirps occurred often during active locomotion (directed either forward or backwards), type 3 chirps seemed to occur more evidently towards the end of a forward-oriented swimming bout, while rises appeared to be centered around forward locomotion (*loc*, *Figure 6A*). Although all chirp types were used during

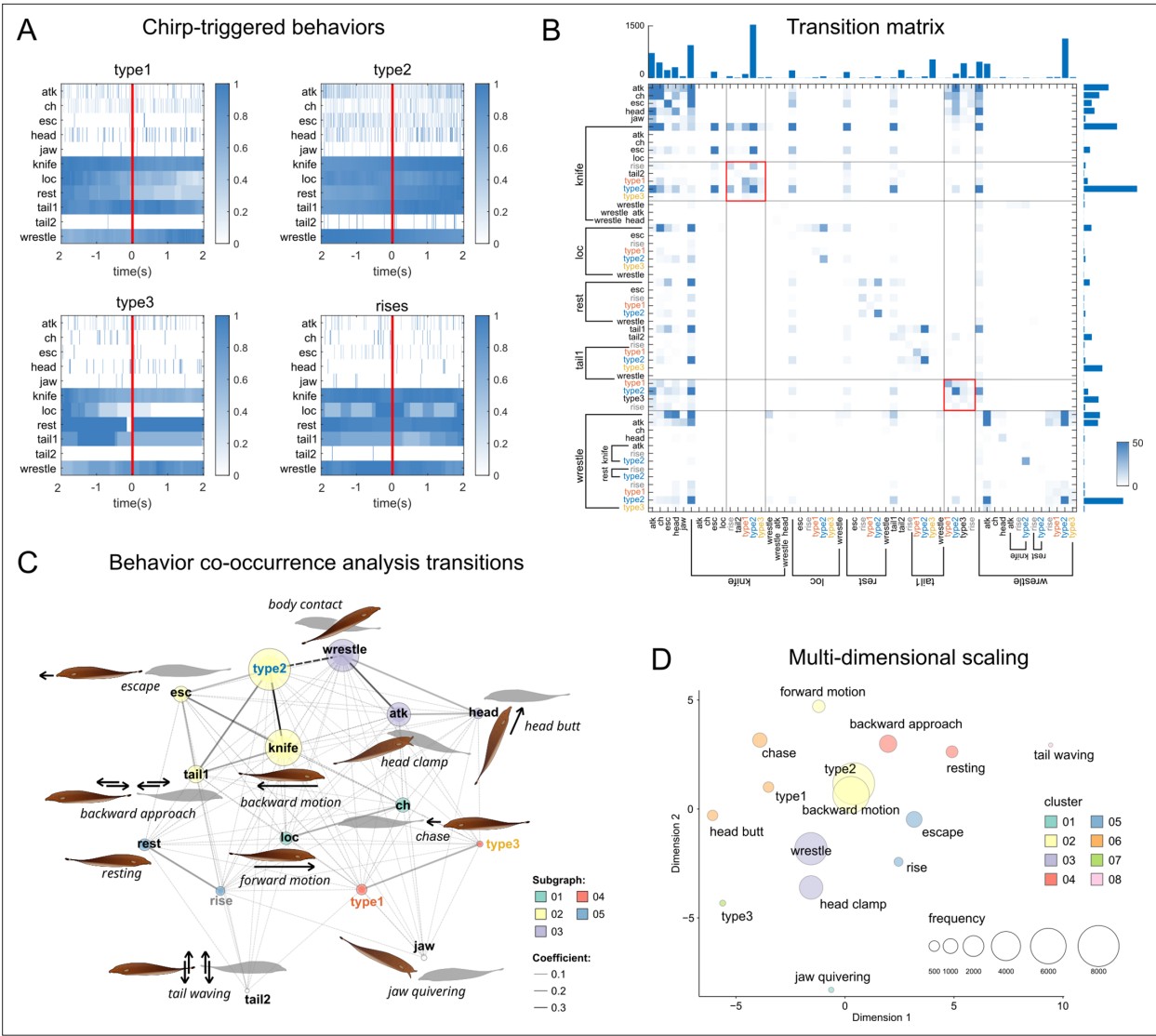

**Figure 6.** Chirping is mainly correlated with locomotion and active sensing. (**A**) Peri-stimulus time histograms (PSTH) centered around different chirp types (window = 4 s) indicating that chirps are reliably emitted during locomotion-related behaviors ('knife', 'loc', and 'tail1') but also during resting states ('rest'). Chirps do not seem to trigger any defensive or aggressive behavior directly (i.e. no evident left/right bias in the PSTH). A more detailed description of each behavior is provided in the Materials and methods (see also panel C). (**B**) Transition matrix showing the total number of behavioral transitions (data pooled from all pairs). Note that all behaviors are considered both individually as well as together with those co-occurring (example: knife-type2 indicates a type 2 chirp produced during backward swimming). The color threshold has been lowered to visualize the most significant transitions. The marginal bar plots represent the sum of the number of transitions along each axis. (**C**) To extract meaningful behavioral correlations, a text-mining algorithm (see Materials and methods) was used to perform a co-occurrence analysis on the whole set of annotations in chronological order (data pooled from 12 fish pairs). In the graph, words closely associated with each other are linked with darker lines (according to a 0-1 coefficient scale). This analysis emphasizes the presence of modularity in a group of strings (i.e. chirps and behaviors), which represents the degree of partitioning of the whole word dataset (N = 5). Each cluster delineates strings sharing similar co-occurrence patterns (see color code). (**D**) The multi-dimensional scaling of the same word-database (2D in this case) shows behavioral combinations having similar patterns of occurrence, regardless of their co-occurrence with other behavioral instances. For this representation, each word is considered individually and the color code represents the clustering of objects based on their proximity (method: Kruskal, distance: Jaccard, N of clusters = 8).

aggressive interactions, these seemed to be rather less frequent in the immediate surround of the chirps (*Figure 6A*).

A tendency for a decrease in attack rates (*atk* and *head*) could be seen perhaps more clearly for type 2 chirps, possibly due to the higher number of this chirp type (89% of all chirps were type 2). Chirps and behaviors occurred with some overlap, depending on whether they represented focal instances or continuous states (such as swimming, wrestling or resting). To assess the presence of

direct correlations between individual and composite behavioral patterns including chirps, aggressive, defensive or explorative behaviors, a transition matrix (TM) was calculated using all recorded transitions (*Figure 6B*).

This allowed us to make the following considerations: (1) as previously observed, type 2 chirps are the most represented chirp type. They are mostly produced in repeats and predominantly during backward swimming (*knife*), they are also followed – to a lower extent – by direct contact (*wrestle*), head bites (*atk*), backward tail-to-tail swimming (*tail1*), escape attempts (*esc*), type 3 chirps, head butts and chasing; (2) behaviors occurring during aggressive encounters (offensive or defensive) are related to one another and also occur in repeats; (3) most chirps occurring during backward swimming are followed by other chirps within the same activity bout, but if attacks are present (*knife* + …+*atk* or+*head*) chirps are not (i.e. attacks may follow chirps but not during backward motion); (4) when used during forward swimming, chirps are often preceding a chasing attempt or a reversal in the swimming direction (*loc → knife*).

While this approach allows to assess chirp-behavior correlations considering even complex combinations, the occurrence of over-represented instances may bias the results by masking potentially significant but less frequent correspondences between complex behavioral patterns. In addition, it has the potential limitation of exaggerating differences among similar behavioral patterns simply because they do not always occur in an identical way.

A way to overcome these issues is to conduct a nearest-neighbor analysis of behavioral sequences, considering not only strictly consecutive instances (as typically done in a transition matrix) but also more general probabilistic correlations among them. To this aim, we have used a *quantitative content analysis* approach using freely available software for text mining (KH Coder; *Higuchi, 2016*; *Figure 6C and D*). Notably, this analysis handles words only from a statistical point of view (i.e. as character strings) and does not evaluate semantic correlations. Data from all behavioral observations were pooled into one single text file, maintaining the original chronological order, subsequently uploaded in KH-C.

We first calculated a co-occurrence network (CON), which linked different strings based on their co-occurrence within the dataset. Importantly, this calculation does not consider only pairwise transitions in which a given word is used but also deals with whole sets of correlated transitions. As a result, words are clustered together based on similarities in the usage pattern of their transition sets (*Higuchi, 2016*).

The resulting co-occurrence diagram confirmed that chirps are used more often in association with swimming activity (*knife*, *loc*, *tail1*, *esc*) and direct physical interaction between fish (*wrestle*). More explicitly aggressive behaviors remained more loosely connected (*Figure 6C*). Interestingly, resting states and rises seem to be used in similar ways (*Figure 6C*) despite the scarce transition probability from one to the other. Conversely, forward swimming (*loc*) and chasing (*ch*) behaviors seemed to co-occur quite often.

To validate these results, we also conducted a factor analysis (multidimensional scaling, MDS) to represent different behaviors on transformed coordinates based on their ranking and usage in the dataset (*Figure 6D*). This complementary approach was used to evaluate more simply the extent of clustering among words having similar appearance patterns, regardless of co-occurrence frequencies. Factor analysis confirmed that type 2 chirps are mostly associated with backward swimming (*knife*) but also with forward motion (*loc*). Escape and rises are clustered together, as they are wrestling and head clamping. Chasing and head butts clustered with type 1 chirps. Type 3, jaw quivering and tail waving did not seem to cluster with any other behavior, probably due to their lower frequency within the dataset.

Overall, these results provided mutually reinforcing evidence indicating that chirps are produced more often during locomotion or scanning-related motor activity and confirm previous reports of a lower occurrence of chirping during more direct aggressive contact (as shown also by *Triefenbach and Zakon, 2008*; *Hupé et al., 2008*). This mainly applies to type 2 chirps and backward swimming (*knife*; *Triefenbach and Zakon, 2008*), while larger chirps – often co-occurring (type 1 and 3 together, *Figure 6C*) – seem to be associated more frequently to fast-occurring behaviors (forward swimming and chasing), but not necessarily in their immediate proximity (as shown by the PSTH plots). Interestingly, despite this co-occurrence, type 3 chirps seem to cluster independently in the factor analysis, possibly suggesting differences in their usage pattern when compared to other large but faster chirps (type 1).

The significantly higher extent of chirping during swimming and locomotion, consistently confirmed by four different approaches (PSTH, TM, CN, MDS), suggests that – although chirp-behavior correlations may exist at time-scales larger than those here considered – chirping may be linked more strongly with scanning and environmental exploration than with a particular motivational state, thus confirming findings from our playback experiments.

## Chirps significantly interfere with the beat and alter the patterning of transcutaneous voltage

Given the stereotypical use of chirp types across various social contexts (*Appendix 1—figure 3*), the lack of chirp type-specific responses (*Appendix 1—figure 4*), the higher degree of auto-correlation of chirp time-series (*Figure 5*), the co-occurrence of chirps with locomotion and scanning (*Figure 6*), and the significant influence of beat frequency (i.e. DF) on chirp variance (*Figure 2B*), we propose that the function of chirps is linked to physiological aspects of beat processing in the chirping fish. According to this hypothesis, the emission probability of different chirp types would depend on how detectable their interference with the beat is, specifically at the beat frequency at which they are produced (for an in-depth analysis see *Walz et al., 2013*; *Walz et al., 2014*). It is worth noting that the frequency-dependent effects of different chirps on the beat have already been extensively described (*Walz et al., 2014*; *Petzold et al., 2016*). To evaluate this possibility, we computed an estimate of the interference caused by chirps on beat periodicity (*beat interference*) using an array of Gaussian chirps across a wide range of beat frequencies (±300 Hz, *Appendix 1—figure 9*). Estimates of beat interference were made by calculating the ratio between the cumulative duration of the beat cycles affected by a given chirp (1 beat cycle corresponding to the interval between two consecutive beat peaks) over the cumulative duration of all the beat cycles within the time window used as a reference (700ms, for other window sizes see *Appendix 1—figure 10*). This ratio was calculated for a wide range of chirp parameters: duration ranged between 10 and 400ms, while FM ranged between 0 and 400 Hz (*Appendix 1—figure 9A*). Since chirps can be produced at any EOD phase (*Walz et al., 2013*), estimates of beat-interference were calculated for each chirp at four different phases (with 90° steps) and then averaged. As a result, we obtained a three-dimensional array of beat-interference values which could be matched to recorded chirps (N=30,486 chirps, from our first set of recordings) having the same parameters (duration, FM and DF; *Appendix 1—figure 9B*).

By plotting both beat-interference values and chirp abundance by DF, we found that emitted chirps were indeed often localized in correspondence with areas of high interference: at negative DFs, chirps were localized mainly in the type 2 range, whereas at positive DFs recorded chirps overlapped more often with the type 1 and 3 ranges (*Appendix 1—figure 9C*). This resembled what was already observed during the playback experiments in which EOD frequency ramps were used (*Figure 4*). Eventually, by calculating the normalized cumulative interference for each chirp type, a high degree of matching was found between the DF values at which the interference is higher and those at which chirps were actually recorded (*Appendix 1—figure 9B*).

A chirp type comparison (*Appendix 1—figure 9D*) further revealed that type 2 chirps at low or even negative DFs may induce similar interference levels as those induced by larger chirps at higher beat frequencies. This agrees with what has been previously reported using other parameters to estimate chirp-induced beat perturbations (i.e. chirp conspicuousness, *Petzold et al., 2016*) and could perhaps explain the lower occurrence of larger chirps in low frequency fish (more often in negative-DF encounters). In addition, areas of lower interference seem to be symmetrically localized around 60 Hz, the frequency level to which electroreceptors are most sensitive (*Bastian and Electrolocation, 1981*) and at which chirp occurrence was lower (*Appendix 1—figure 9B–D*). However, this could be also due to the lower representation of these DFs values in this sample since responses to frequency ramps do not seem to convincingly validate this idea (*Figure 4*). Overall, these results corroborate the hypothesis that chirp production may be explained mainly by the effects chirps have on the beat perceived by the chirping fish (as emphasized by *Walz et al., 2014*).

Differences in the beat interference elicited by different types of chirps were found: the interference induced by type 1 chirps is almost one order of magnitude higher than the estimates of the same parameter for type 2 chirps, but one order of magnitude lower than the beat interference induced by type 3 chirps. Rises induce interferences in the range of type 1 chirps (not shown).

Obviously, measuring chirp-triggered beat interferences by using an elementary outlier detection algorithm on the distribution of beat cycles does not reflect any physiological process carried out by the electrosensory system and can be therefore used only as an oversimplified estimate. A more realistic representation of beat processing and chirp-triggered beat perturbations can be done through electric images (EIs), which are 2D representations of the current flowing through electroreceptors located on the fish skin at any given moment in time (*Kelly et al., 2008*). In the context of social interactions, EIs represent the projection of the electric field generated by two interacting fish onto the array of electrosensory receptors, at the net of the environmental perturbations caused by objects, motion and other sources of noise (*Kelly et al., 2008*; *Fotowat et al., 2013*).

Both, the 3D shape of the electroreceptor array (i.e. the shape of the fish body) and the asymmetric geometry of the electric field contribute to the further alteration of electric images during locomotion (*Rasnow and Bower, 1996*; *Yu et al., 2012*). While the EOD is one order of magnitude stronger at the tail end compared to the head, the EOD harmonic components also change along the body axes. This means that the perception of beat contrast can change at different body locations (*Rasnow and Bower, 1996*). This explains also why electric images are easily altered by motion and why, in the presence of anisotropic electric field diffusion caused by soluble materials or by environmental objects, beat perception is not an homogeneous process throughout the whole population of electroreceptors. In fact, depending on the fish relative positioning and orientation, electric images can be subject to significant degradation or near cancellation, contingent on the phase of interacting EODs (*Kelly et al., 2008*). To prevent this, image contrast can, in principle, be enhanced either by adjustments of the body position or through other mechanisms amplifying the responses of either electroreceptor afferents (P-units) or downstream circuits (ELL pyramidal cells). Since the beat AMs generated by the chirps always trigger reliable responses in primary electrosensory circuits (pyramidal cells in the ELL respond to both increases and decreases in beat AM), any chirp-triggered AM causing a sudden change in P-unit firing could potentially amplify the downstream signal (*Marsat and Maler, 2010*) and thus enhance EI contrast. If this is the case, chirps should be able to trigger significant EI changes at different body locations. Obviously, if such effects are visible in simplified static scenarios (e.g. measuring transcutaneous voltage resulting from two static EODs at different body locations), they should be even more evident in situations in which the interacting fish are moving around each other.

To address this point, we first assessed the impact of chirps on EIs by computing EIs using theoretical chirp examples and 778 Hz as carrier EOD frequency (*Figure 7A and B*). Chirps were here defined as Gaussian frequency changes with the following parameters: type 1=200 Hz, 40ms; type 2=100 Hz, 20ms; type 3=300 Hz, 100ms; rises = 50 Hz, 100ms. For each chirp type, a signal amplitude loss – similar to that occurring in natural chirps – was applied: type 1: 47%, type 2: 7%, type 3: 45%, rises: 3% (*Engler et al., 2000*; see also *Figure 1E*). The receiver frequency was set at different values expressed in percentage of the sender's EOD frequency: from –20 to +20%, with a 5% step. The resulting beats were phase-shifted using 45° angular steps.

We then assessed the combined effects of chirps and reciprocal positioning on EI (*Figure 7C*). Comparisons of chirp-EI with beat-EI (without chirps) consistently showed that chirps led to increased changes in the voltage measured as area under the curve (AUC) of the beat envelope (*Figure 7D*).

This means that chirp-triggered electric images can potentially convey spatially relevant information, detectable by integrating the EI input resolved by electroreceptors at different body locations.

We next quantified this effect including more reciprocal locations around the sender fish (randomly selected, *Figure 7C and E*). Comparison of chirping beats with constant frequency beats showed a significant effect of chirps in both sender and receiver fish (chirp vs beat, black asterisks in *Figure 7E*). This effect is often more evident for sender fish (sender vs receiver, red asterisks in *Figure 7E*).

This indicates that chirps consistently alter beats and EIs, at all the different reciprocal positions considered, thus supporting the idea that chirp-triggered EIs contain usable spatial information. Overall, both the estimates of chirp-induced beat interference as well as the more realistic representations of chirp-triggered electric images, confirm that chirping exerts conspicuous and spatially referenced effects on the beat that should be easily detectable by electrosensory afferents and could therefore be usable to complement beat processing, perhaps in scenarios in which beat amplitude is either low or poorly resolved.

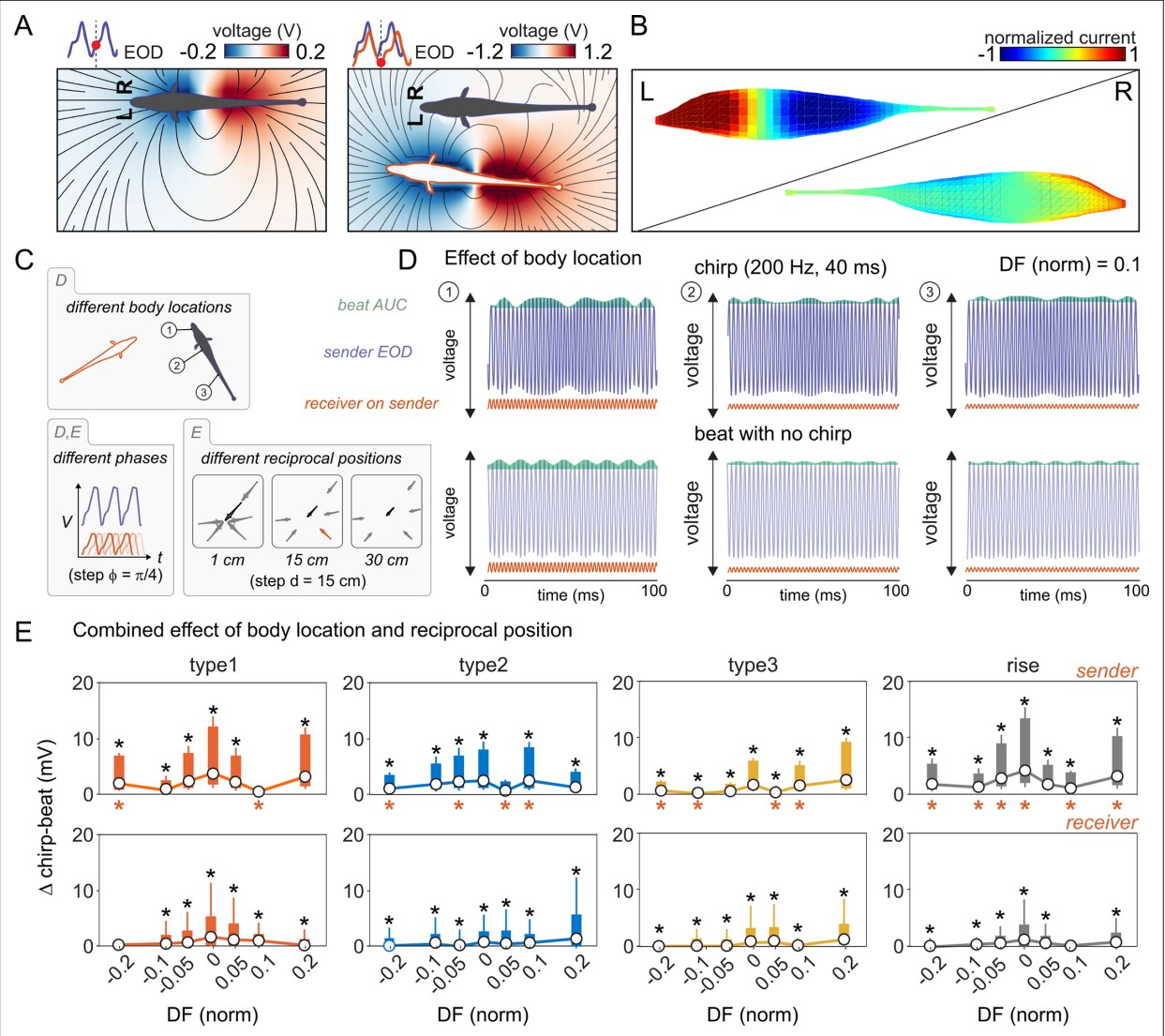

**Figure 7.** Chirps alter estimates of transcutaneous voltage in sender and receiver fish. (**A**) Heatmaps representing the electric field (V/cm) generated by a sender fish alone (left) and by two interacting fish (right; fish length = 15 cm, distance = 10 cm). The electric field lines induced by the sender fish's EOD (gray silhouette) and that of the receiver conspecific (white silhouette) are shown in black. (**B**) Normalized heatmaps superimposed on the sender fish's contour at the same instant as in A, representing the boundary element model (BEM, see Materials and methods) simulation of the electric image resulting from field interactions measured on the sender fish's skin. (**C**) Top: Schematic example of two fish, a chirp sender (black fish) and a chirp receiver (red contour), and the location of 3 body points used in D. Bottom left: all simulations include 8 different EOD phases (in 45° steps). Bottom right: the simulations used in E, include different body locations and are calculated across 835 nodes for each fish. In addition, different reciprocal body positions are included (i.e. distances and angles) for the calculation of the electric image data in E. (**D**) Voltage measured at three points on the sender fish's body as shown in C (upper panel) for a baseline 10% difference in fish EODf (i.e. normalized DF = 0.1). Top row: chirp condition, bottom row: beat with no chirp. The green area represents the integral of the voltage change over time (i.e. beat area under the curve, or AUC). The red signal indicates the chirp receiver's EOD at the same three points on the chirp sender's body. (**E**) Net effect of chirps on electric images for different fish orientations and distances (insets in C, bottom right), represented as the sum of the voltage integral (AUC) over time (measured throughout the beat) due to chirps across the entire body (835 nodes), compared to the carrier beats alone, for seven baseline differences in fish EODf. Black asterisks indicate significant chirp-beat differences. Data for the sender fish (top) and receiver fish (bottom) are displayed separately. Significant differences between sender and receiver are indicated with red asterisks.

## Chirping is significantly affected by locomotion and social distance

So far, our analysis focused on interacting fish pairs within a relatively small tank environment (35x80 x 35 cm). In such a scenario, we can expect fish to be able to detect the beat caused by the partner's EOD at almost all times. As we pointed out in the previous paragraph, chirping could potentially convey enhanced spatial information even at short distances.

To obtain more insight into how chirps are used in more naturalistic scenarios, we evaluated chirping and different aspects of swimming activity using a *novel environment exploration* assay involving a larger recording aquarium (160x80 x 20 cm) containing shelters (plastic tubes and plants), barriers and other objects with which the animal had no previous experience. We assessed chirp production by brown ghosts freely swimming while localizing a beat source (a caged conspecific). This allowed to evaluate aspects of fish locomotion and social behavior neglected in previous experiments, potentially explaining high inter-individual chirping variability (e.g. thigmotaxis, novelty-seeking, sociability). If chirping relates to geometrical aspects of beat processing (distance, orientation, direction, and motion), a bias for any of these should be evident in these experiments. If chirps enhance beat processing and conspecific electrolocation, for instance, chirping should occur within beat detection range but at a certain distance. Alternatively, if chirps help resolve other parameters of electric field geometry, they may be produced at specific angles or when fish are in close proximity.

The caged conspecific was located in a mesh tube on one side of the arena (*Figure 8A*). A 'novel object' (a 1x5 cm graphite rod) was hidden behind a barrier and used to test novelty seeking. Different regions of interest (ROIs) were used to score the occurrence of fish swimming behavior around the shelter area, the tank walls (within 5 cm distance), the novel object and the conspecific (*Figure 8A*). The time spent swimming in the arena's open space (5 cm away from the walls) was also evaluated. Behavioral observations were conducted considering the subject's sex, as it is the behavioral variable most directly correlated to DF and beat frequency.

We found that on average brown ghosts spent little time exploring the caged conspecific (12% of the time) and the shelter (14%), while slightly more attention was paid to the novel object (18%), swimming along the walls (ca. 20%) or exploring other objects in the 'open' environment (ca. 35% of the time). In particular, female brown ghosts spent more time swimming in the open space areas of the arena, as opposed to swimming closer to the tank walls (thigmotaxis; *Figure 8B*). Female subjects also spent more time investigating the novel object than the conspecific (*Figure 8B*). Male behavior showed a different trend as males spent comparatively more time swimming close to the tank walls and more time in contact with conspecifics (*Figure 8B*). Possibly as a direct consequence of this, they also produced a higher number of chirps, compared to their female counterparts. Most chirps were produced in close proximity to the caged conspecific (*Figure 8C*). Analysis of the angles and distances of fish during chirps revealed that first, chirps were mostly produced when subjects were located at 30°, 150°, or 330° angles relative to the axis defined by the mesh tube (*Figure 8D*), and second, most chirps were produced at a distance of less than 25–50 cm (*Figure 8E*). While the angle bias could be due to the circular loops described by fish swimming around the conspecific, the rather limited distance range at which chirps are produced could be explained by the range of the EOD field (beat amplitude reaches 1% at around 30 cm; *Fotowat et al., 2013*; *Henninger et al., 2018*; *Benda, 2020*; see Appendix 1) and confirms previous observations (*Hupé et al., 2008*; *Zupanc et al., 2006*; *Henninger et al., 2018*).

Interestingly, chirps of different types were not all produced at similar angles: while type 1 and type 2 chirps seem to be more often produced at 30–150°, type 3 chirps – produced in this experiment only by males – seem to be used in the ±30° range (i.e. on the right side of both the arena and the plane representing the angles). Rises – in both males and females – seemed more often to be produced at multiples of 30° located on the left side of the cartesian plane (30°, 150°, and 210°). All chirp types, except for rises, were produced within a 50 cm radius from the conspecific fish (*Figure 8E*). Rises were produced almost ubiquitously, by both sexes, and most of the time at locations far away from the caged fish (*Appendix 1—figure 11*).

Factor analysis (FAMD) confirmed the already known sex-dimorphisms in chirp distribution (males chirp more than females and more often towards other males, *Appendix 1—figure 12*). Collectively, caged fish produced more chirps than freely swimming subjects (4410 vs 3483 chirps). In addition, variables associated to beat frequency (such as DF, EOD frequency, sex, chirp type) explain a good percentage of chirp variance (73.7% of variance explained as opposed to the 55% explained including all other variables 55%; see also *Figure 2B*). A relatively high loading on principal components was also found for variables associated with locomotion, such as distance and time spent exploring different tank areas (*Appendix 1—figure 12B*). Accordingly, chirp clustering is affected by the EOD frequency difference of the emitting fish (DF) and by distance (*Appendix 1—figure 12C*, insets).

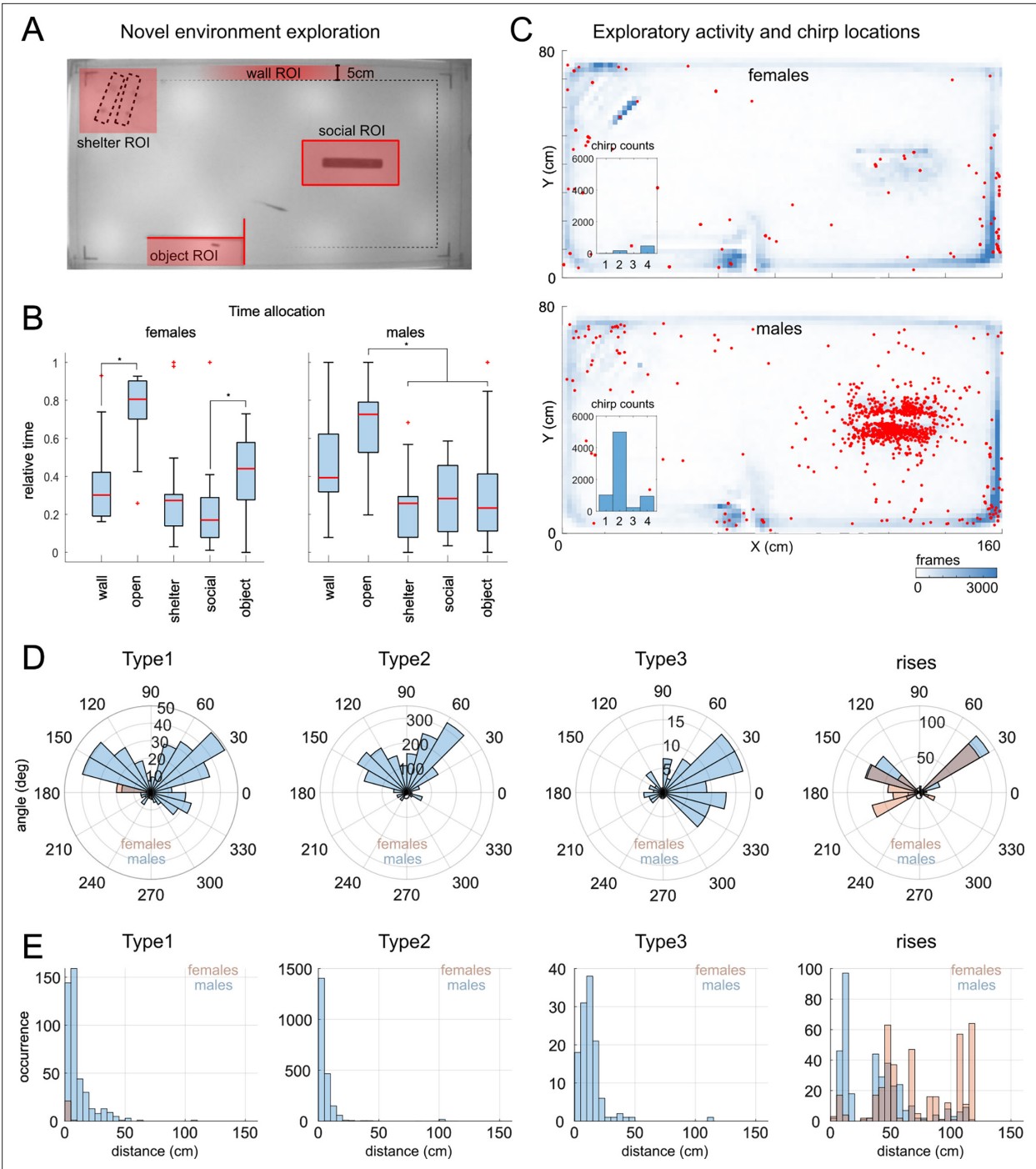

**Figure 8.** Chirping during novel environment exploration. (**A**) Diagram of the recording arena showing the criteria used to define the different regions of interest (ROIs): (1) the presence of shelters (PVC tubes), (2) the proximity to the tank walls (distance < 5 cm), (3) the presence of a fish (caged in a mesh tube), (4) the presence of an 'unknown/novel' conductive object (a 3 cm piece of graphite). (**B**) Proportion of time spent in the different ROIs (N = 14 females, Friedman $X^2$ 22,9 p < 0.001 wall vs open p < 0.001, social vs object p = 0.02, open vs shelter/social/object < 0.01; N = 15 males, Friedman $X^2$ 19,8 p < 0.001 wall vs open p = 0.049, wall vs shelter/object p < 0.05, open vs shelter/social/object p < 0.001). (**C**) Chirp locations (red) overlaid to the heatmaps showing the average swimming activity of females and males (for the specific locations of different types of chirps see **Appendix 1—figure 11**). (**D**) Polar histograms showing the angles between the two fish during chirping. Angles are referred to the X-axis and are sorted based on chirp type and sex (male = blue, female = red). (**E**) Histograms of the chirping distances in males (blue) and females (red) relative to different types of chirps.

Modelling of field current and iso-potential lines using dipole models of our fish allows to visualize more intuitively how both chirp angles and distances can be explained by the electric field geometry (*Appendix 1—figure 13A*). Most chirps (90%) are in fact produced within a distance corresponding to 1% of the maximum field intensity (i.e. roughly 30 cm; *Appendix 1—figure 13B*), indicating that chirping occurs way above the threshold distance for beat detection (i.e. roughly in the range of 60–120 cm, depending on the study; see appendix 1: *Detecting beats at a distance*). This effect is even more evident when chirp locations are compared to the values of field intensity measured at different distances (*Appendix 1—figure 13C, D*). By estimating the interference each chirp type would induce on the beat, based on its decay law (*Appendix 1—figure 13D*), one can easily see how the two are linearly and directly correlated (*Appendix 1—figure 13E–H*).

Overall, given that most chirps are reliably produced within very short range, they may not be useful for improving beat detection per se. Conversely, the orientation bias for chirp locations suggests that chirps are used during specific swimming trajectories and thus likely occurring while brown ghosts are trying to resolve other spatial parameters, such as the orientation, direction and possible motion of the beat source. In all this, rises may represent an exception as their locations are spread over larger distances and even in presence of obstacles potentially occluding the beat source (i.e. shelters, plants, or walls), all conditions in which beat detection or beat processing could be more difficult or simply interrupted (an idea coherent with the production of rises right at the end of EOD playbacks; *Appendix 1—figure 6*). Taken together, these observations may possibly explain why female subjects, apparently less motivated in exploring social stimuli, also produce considerably fewer chirps unless they are forced into social interactions (*Appendix 1—figure 2E, F*). More broadly, these findings indicate that chirping, besides being strongly correlated to the frequency of the beat generated by the interacting fish, is also significantly affected by locomotion and by spatial relationships between subjects within beat detection range.

## Chirping increases with environmental complexity

We hypothesized that if chirps are used as probes to refine beat processing and/or the resolution of electric images produced by conspecifics, higher levels of environmental complexity would likely result in higher chirping rates (as it implies higher probing needs, *Siemers et al., 2009*). To test this idea, we used both playback experiments and recordings of interacting fish pairs. The objective was to evaluate the chirping responses of brown ghosts to EODs in presence of environmental clutter which would distort electric fields in tangible but unpredictable ways (*von der Emde, 2006*).

We first recorded social interactions using the same divided tank configuration as in our previous recordings. Fish pairs (N=6) were recorded in 3 different conditions: (1) with lights turned ON and no object in the tank other than the mesh divider placed between the 2 fish (lights ON - clear), (2) with lights switched OFF (lights OFF - clear) and (3) with lights OFF and the addition of shelter PVC tubes and plastic plants filling almost entirely the two tank compartments (lights OFF – cluttered; *Figure 9A*). Fish pairs were recorded for 15 min with no previous knowledge of either the environment or their social partner. Trials were shuffled in a randomized order and were spaced by 5 min inter-trial intervals. Chirp counts were evaluated at the end of the trials. As expected, the total chirp rate increased in absence of light (*Zupanc et al., 2001*). More strikingly, the increase was even more pronounced when clutter was present in the environment (*Figure 9B*). In all cases, type 2 chirps were the most often produced type (*Figure 9C*). A slight (non-significant) increase in type 2 and type 3 chirps could be observed under cluttered conditions. These results clearly indicate that chirping is affected by environmental complexity, since in these recordings (1) aggression- or courtship-related behaviors were either absent or randomly occurring in the three conditions and (2) both fishes had equal environmental and social experience.

Nonetheless, even in conditions of low 'electrical detectability', fish can sense each other using visual cues, olfactory and mechanical cues induced by water movements during swimming. Therefore, chirp production in these experiments could also be affected by a range of sensory inputs co-occurring in an unnaturally confined space and for the whole duration of the recordings (15 min). To rule out these possibilities, we conducted playback experiments in a larger enclosure (160x50 cm), allowing fish to detect playback EODs from a wider range of distances. Playback electrodes were not directly accessible but placed behind a mesh barrier located at either side of the enclosure (*Figure 9D*). Environmental clutter (plastic plants) was interposed between the electrodes and the mesh divider at

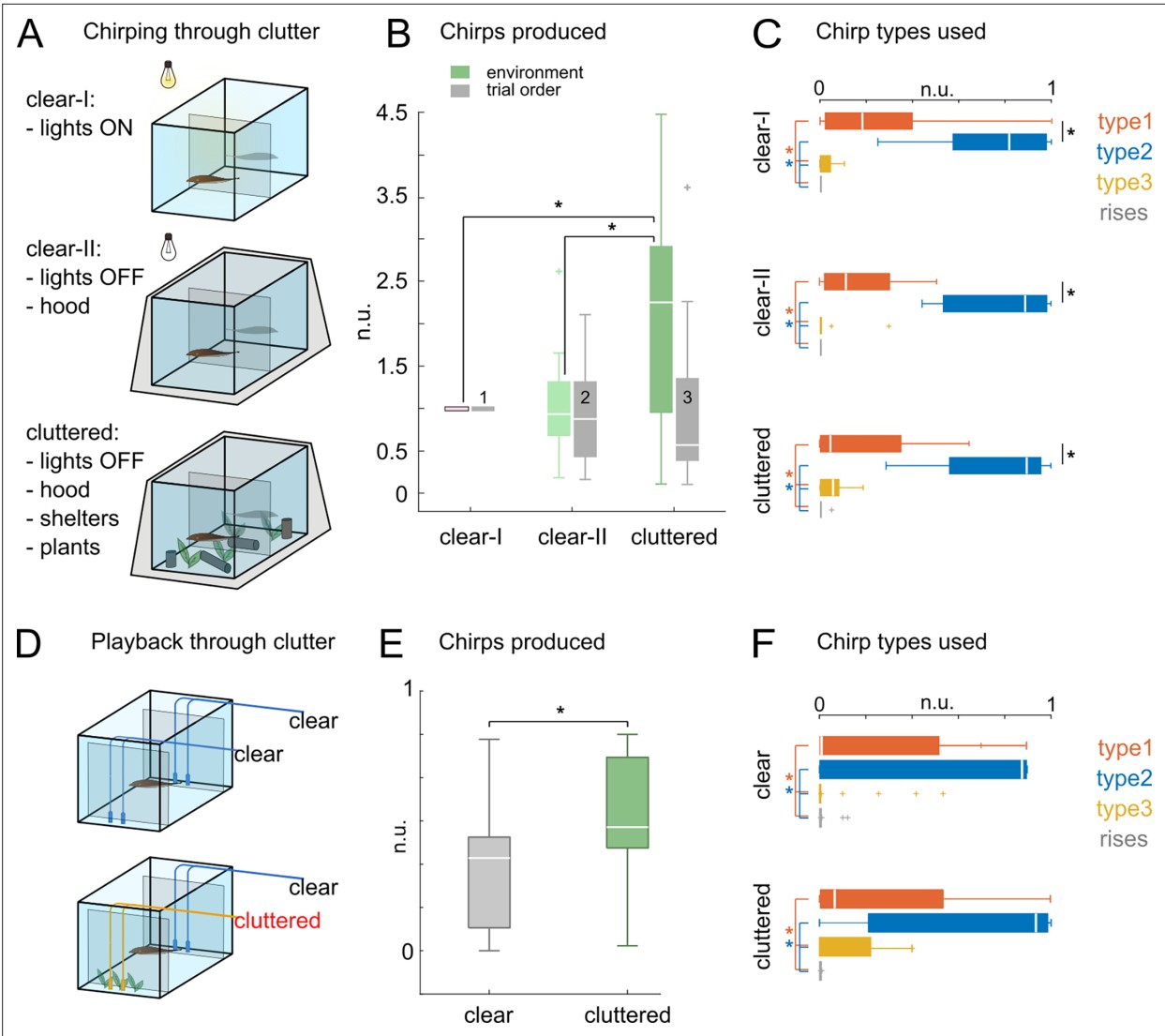

**Figure 9.** Effect of environmental clutter on interacting fish pairs. (**A**) Recording of fish pairs (N = 6) in environments of different sensory complexity: lights ON = clear tank environment and direct illumination, lights OFF = clear tank environment, no illumination, lights OFF + clutter = no illumination and cluttered environment. (**B**) Total amount of chirps produced in each condition and normalized on the lights ON session (green bars; Friedman's p = 0.053; lights ON vs lights OFF + clutter = 0.024, lights OFF vs lights OFF + clutter = 0.051). Chirp counts relative to trials sorted in chronological order (1-3) are shown in gray. (**C**) The box plots show the normalized chirp type counts relative to each session (lights ON, Friedman's $X^2$ = 21.9 p < 0.001, pairwise comparisons 1 vs 2 p = 0.006, type 1 vs 3, type 1 vs 4, type 2 vs 3, type 2 vs 4 p < 0.001; lights OFF Friedman's $X^2$ = 21.8 p < 0.001 pairwise comparisons type 1 vs 2, type 1 vs 3, type 1 vs 4, type 2 vs 3, type 2 vs 4 p < 0.001; clutter Friedman's $X^2$ = 19.5 p < 0.001 pairwise comparisons type 1 vs 2, type 1 vs 4, type 2 vs 3, type 2 vs 4 p < 0.001). (**D**) Results of playback experiments in which EOD mimics were either directly detectable through a fine mesh barrier (clear) or more indirectly due to a barrier of plastic plants interposed between the mesh and the EOD source (cluttered). Clear and cluttered trials were presented in random succession (N fish = 6, 10 trials each, 60 sec ITI). (**E**) Total chirp counts in the two conditions are normalized on the total amount of chirps produced by each subject (Wilcoxon, p=0.025). (**F**) Boxplots showing the chirp type composition of each condition (clear Friedman's $X^2$=17.4 p<0.001 pairwise comparisons type 1 vs 3 p=0.048, type 1 vs 4, type 2 vs 3, type 2 vs 4 p<0.001; cluttered Friedman's $X^2$=29.9 p<0.001 pairwise comparisons type 1 vs 3, type 1 vs 4, type 2 vs 3, type 2 vs 4 p<0.001; Wilcoxon type 1 clear vs type 1 cluttered p=0.034). Tank sizes: A-C: 30x80 cm; D-F: 160x50 cm.

a random location in each trial. We found that chirp rates increased in response to playback signals specifically when they were partially concealed by clutter (*Figure 9E*). Notably, this effect seems to be more evident for larger chirps (type 1 and type 3; *Figure 9F*), although combined effects of chirp types and clutter have not been found. Overall, these results indicate that chirping is significantly affected by the presence of environmental clutter partially disrupting – or simply obstructing – the processing of beat related information during locomotion. They further confirm electric signals alone

are sufficient to trigger the chirping increase in cluttered conditions. All chirp types seem to be equally affected, with type 2 chirps being the most common type produced also in cluttered conditions.

The increased utilization of chirping during locomotion around partially concealed EOD sources mirrors the higher probing rates typically observed in complex environments (*Obrist, 1995*; *Schnitzler and Kalko, 2001*; *Slabbekoorn and Peet, 2003*; *Siemers et al., 2009*). This supports the notion of chirps as self-referenced probing cues, potentially employed to optimize short-range aspects of conspecific electrolocation, such as size, orientation, and direction – a hypothesis that will certainly be explored in future studies.

## Discussion

In this study, we examined the factors influencing the production of EOD frequency modulations (chirps) in brown ghost knifefish. We gathered a substantial dataset of chirps recorded during various social interactions, evaluating the impact of territory ownership, social hierarchy, sex, reproductive state, and social experiences. Analyzing this data, we found that the most influential factors in chirping are primarily beat frequency (DF) and experience (both environmental and social). Chirp patterns mainly consist of repetitions of the same chirp type which are heavily influenced by DF. Responses to playback chirps inserted into a sinewave stimulus of a particular DF are indistinguishable in terms of chirps emitted by the fish and exploratory behavior (approach/avoidance). Our behavioral observations of freely swimming fish indicate that chirping typically occurs at short ranges, often coinciding with active social approaches. Building upon these findings, we suggest that chirps serve as self-directed signals (sensory probes), aiding in the processing of spatial information derived from beats.

### Existing functional hypotheses

Since the earliest reports (*Larimer and MacDonald, 1968*), chirps have been assigned behavioral meaning as communication signals based on the following observations (mostly in brown ghosts): (a) fish chirp mainly during social encounters; (b) males chirp considerably more than females (*Dye, 1987*; *Bastian et al., 2001*; *Tallarovic and Zakon, 2002*; *Dunlap and Larkins-Ford, 2003*; *Kolodziejski et al., 2007*) and are also considerably more aggressive; (c) during male interactions (often agonistic) smaller chirps are produced (i.e. shorter duration and lower frequency excursion: type 2) whereas larger chirps (types 1 and 3) are used more often during female-male encounters (presumably courtship-driven); (d) high chirp rates are observed in mating pairs, too, with a relatively higher production of large chirps (i.e. type 1 and type 3; *Hagedorn and Heiligenberg, 1985*; *Henninger et al., 2018*). This led to a dualistic semantic subdivision of chirps, with small chirps implied in agonistic interactions and larger chirps pertaining to different aspects of courtship and mating.

Although intuitive, this idea raises the following concerns: first, several factors (sex and the magnitude of beat-interference) covary with the main corollary of chirping, that is beat frequency. These covariates are not easy to disentangle, which makes it difficult to separate neuroendocrine or other behavioral factors from the biophysical features of EOD fields affecting detection and localization of conspecific fish. Second, observations in a few species are generalized to all other gymnotiforms without testing whether chirping may have similar functions in other species (*Turner et al., 2007*; *Smith, 2013*; *Petzold et al., 2016*). Third, social encounters do not only involve communication but also higher levels of locomotion and reciprocal exploration, which could also be linked to chirping activity, but have not been considered in the past. Fourth, studies supporting a dual function of chirping (i.e. aggression and courtship), only rarely addressed this hypothesis using breeding animals. Last, and most importantly, no causal evidence in support of any of the hypothesized functions of the different types of chirps has ever been provided.

The idea of chirps being used as *probes*, as proposed here, is not entirely new: due to the tight correlation between chirp features and beat frequency (mainly chirp frequency modulation), Walz and coauthors previously hypothesized a role of chirps in determining the sign of the beat frequency (*Walz et al., 2014*). This indeed may represent an ambiguous parameter as two opposite frequency differences (or DFs) result in the same beat frequency: for example consider a fish producing an EOD of 750 Hz (EOD1) and interacting with conspecifics with an EOD of either 700 Hz (EOD2) or 800 Hz (EOD3). The two beats could be indistinguishable (assuming – perhaps not realistically – that the same mechanism involved in DF discrimination at lower DF values would not work in this case; *Metzner,*

*1999*). However, since any frequency modulation of EOD1 will have different effects on the beat frequency relative to EOD2 and EOD3, chirps could indeed solve the problem of disambiguating the sign of DF (which in brown ghosts could mean no less than discriminating between male and female conspecifics). However, it is also possible to obtain beat frequency information more reliably with a more gradual and slow change in the fish's own EOD frequency (which knifefish are capable of producing, see for instance *Rose and Canfield, 1991*). More banally, the average chirp rates do not seem to realistically reflect such an aim (at low beat frequencies, it is not rare for a fish to chirp more than 100 times per minute).

More recent studies have further suggested that patterns of specific chirp types could be a salient signal used for communication purposes in the context of mating (*Henninger et al., 2018*). Yet, results obtained from our recordings do not support the existence of a chirp-based syntax in social interactions in which breeding conditions were simulated: fish pairs acclimated to breeding conditions (but not mating) used non-patterned chirp repertoires similar to fish pairs recorded in non-breeding conditions. One exception to the *monotony* of the chirping repertoires observed in this study is given by male-female pairs: in this case, type 1 and type 3 chirps are used more often, while the relative amount of type 2 chirps is lower (*Figure 5*, *Appendix 1—figure 2C, D*). However, this effect is likely to be a result of the combined effect of the pronounced dependency of chirp type on DF together with the higher chirp rates observed in males (i.e. if chirp variety increases with DFs and males chirp more, chirp type variety will be higher when males interact with high beat frequencies). Moreover, the somewhat stereotypical transition between large and small chirps that can be observed during repeated exposure to playback frequency ramps, appears a convincing argument against the *communication-hypothesis*. Interestingly, a recent study on pulse-type gymotiforms suggested that chirps could be used for jamming avoidance. In this case, chirps are not an increase of EOD cycles on a sine wave signal, but an increase in the emitted pulses. The hypothesis was based on the temporal correlations of chirps emitted by interacting fish and the effect chirps exerts on EOD phase precession (i.e. a progressive phase shift between two signals, *Field et al., 2019*). In both cases (wave- and pulse-type chirps), chirps induce abrupt EOD phase changes (for pulse-type see *Field et al., 2019*; for wave-type see *Stöckl et al., 2014*; *Walz et al., 2014*) which may be beneficial to avoid jamming potentially occurring at any given phase. Although the EOD waveform generated by wave- and pulse-type gymotiforms is essentially different (see for an overview *Zupanc and Bullock, 2005*), the similarities between the chirp waveforms (i.e. the instantaneous EOD frequency) may not be a matter of chance (*Field et al., 2019*) and could indicate similarities in the neural mechanisms responsible for their generation.

## Inconsistencies between behavior and hypothesized signal meaning

In our playback experiments we observed no differences in the responses of fish of the two sexes to playback stimuli, regardless of whether they contained a particular chirp type or not: test subjects chirped at comparable rates in response to all types of EOD mimics (containing type 1 chirps, type 2 chirps or rises), confirming that the main factor determining type choice is DF and therefore self-referenced sensory requirements (as opposed to different semantic contents). This idea is further corroborated by the known symmetric distribution of chirp types within the ±400 Hz DF range (*Bastian et al., 2001*; *Engler and Zupanc, 2001*; *Triefenbach and Zakon, 2008*; *Kolodziejski et al., 2007*). Nonetheless, it's plausible that playback stimuli, as employed in our study and others, may not faithfully replicate natural signals, thus potentially influencing the reliability of the observed behaviors. However, EOD waveform and harmonic components – which make natural EODs different from playback mimics – do not seem to play a role in EOD discrimination as EOD frequency alone does (*Dunlap and Larkins-Ford, 2003*; *Fugère and Krahe, 2010*). Moreover, artificial EODs are widely used in electrophysiological studies as they elicit comparable responses to natural stimuli in the electrosensory system (e.g. *Benda et al., 2006*; *Marsat et al., 2009*). Future studies might consider replicating these findings using either natural signals or improved mimics, which could include harmonic components (excluded in this study).

Based on our beat interference estimates, we propose that the occurrence of the different types of chirps at more positive DFs (such as in male-to-female chirping) may be explained by their different effect on the beat (*Appendix 1—figure 9*; *Benda et al., 2006*; *Walz et al., 2013*).

In line with our behavioral data, electrophysiological recordings conducted at all main nodes of the electrosensory pathway did not show consistent chirp-type-specific responses in either peripheral or central brain areas (*Metzen and Chacron, 2017*; *Allen and Marsat, 2018*; *Metzen et al., 2020*). Instead, the temporal correlations between chirps and other events (locomotion-related) seem to be more salient factors, as recently proposed for midbrain circuits whose phasic responses to moving objects are used to timestamp their occurrence during spatial navigation (*Wallach et al., 2018*). Recordings from interacting fish pairs confirmed the absence of any significant correlation between chirp type choice and *behavioral context*, except for those cases characterized by higher beat frequencies (*Appendix 1—figure 3*). This suggests that the effect of *behavioral context* highlighted in our factor analysis (*Figure 2*) is mainly due to the number of chirps produced (*Appendix 1—figure 2*), rather than their type (*Appendix 1—figure 3*).

Type 1 and type 3 chirps (i.e. large chirps), have been often suggested to be courtship related signals because they are emitted predominantly by opposite sex pairs (i.e. fish pairs characterized by a higher DF value; *Hagedorn and Heiligenberg, 1985*; *Henninger et al., 2018*). However, the symmetric distribution of their production probability around the lowest DF values (relative to the sender EOD frequency), suggests that this conclusion may not be always valid. In addition, here we show not only that large chirps are also consistently produced at very low DFs but also that the relative proportion of different chirp types does not seem to differ in social pairings implying potentially very different internal states in the interacting fish (breeding, aggression, territorial competition, etc.). Both results are further weakening the link between large chirps and courtship displays.

Type 2 chirps are also considered to be significantly correlated with aggressive encounters, although it is not clear whether they represent purely aggressive, possibly deterring, signals (*Zupanc, 2002*; *Triefenbach and Zakon, 2008*) or attempts to de-escalate aggression and communicate submission (*Hupé et al., 2008*). Our results do not support either of these hypotheses: type 2 chirps are produced at higher rates at low beat frequencies (a feature of any encounter in which fish with similar EOD frequency approach each other and which has nothing to do with their intentions), they are more often produced by newly introduced fish (which are not necessarily more aggressive but could just be more explorative), less in experienced fish (which also move less; *Appendix 1—figure 2A, D*, *Appendix 1—figure 3*), more by males when separated from females (males are more motivated to interact socially; *Figure 8*) and more in females when freely swimming (females are more motivated to escape when exposed to males directly).

Notably, rises reliably occurred at the end of the playback stimulations (as also reported by *Kolodziejski et al., 2007*), which could be explained by their use as probes to assess signal presence in case of abrupt interruptions or weakening of EOD mimics (either caused by objects occluding temporarily the EOD field – clutter – or by playback termination). Interestingly, rises were considered by other authors as 'proximity signals', although most likely with a different connotation (*Hupé et al., 2008*). Rises larger than those here considered (perhaps more similar to the 'gradual frequency rises' reported by *Zupanc et al., 2001*) were recently associated with chasing attempts (15% of the times) and physical contact (less than 5% of the times; Raab T, personal communication). However, these figures appear too small to allow reliable conclusions about the meaning potentially associated with these frequency modulations. Previous experiments reported the use of long rises (larger events compared to those we have labeled as rises) during agonistic interactions (*Hupé et al., 2008*; *Raab et al., 2021*). However, it is not clear whether these are aggressive signals (as proposed by *Hupé et al., 2008*; *Raab et al., 2021*) or de-escalating submissive signals (as proposed by *Hopkins, 1974*; *Serrano-Fernández, 2003*). In our recordings rises are produced more or less with equal probability by female and male subjects (42.5% of all rises are produced by males, 57.5% by females) and yet, compared to other chirp types, their relative probability is higher in females (1.2% of all chirps in females vs 0.12% in males), although female brown ghosts are less frequently engaged in aggressive interactions.

Taken together, these considerations strongly indicate that the primary determinants of chirp rates are the locomotor activity levels and the necessity to locate other fish through the beat. In our experiments, the choice of chirp type appeared largely unaffected by the behavioral context in which chirps were recorded, while the degree of social interaction, as observed in comparisons between novel and experienced fish pairs, played a bigger role.

## Probing with chirps

In this study we explore the possibility that chirps could improve the processing of spatial parameters associated to conspecific localization by enhancing beat processing. The frequent use of chirping during close physical interactions, suggests that whatever role chirps may play in beat processing, this may not necessarily be related to the mere localization of a beat source at a distance (such as most echolocation calls) but perhaps to other features detectable at close-range (such as size, orientation, direction, and motion). This could be achieved in at least three different ways. First, chirping could temporarily adjust beat frequency to levels better detectable by the electrosensory system (*Bastian et al., 2001*). Second, chirps could be used to improve active electrolocation by briefly enhancing beat temporal resolution (all types of chirps will increase ELL pyramidal cell firing rates, acting on different input lines; *Marsat and Maler, 2010*). Third, as previously discussed within this study, chirps could be used to enhance electrosensory responses to the beat. This may be achieved by simply enhancing the output of the primary electrosensory afferents (through synchronization of different types of input; *Benda et al., 2006*) or by phase-shifting the carrier EOD in correspondence of the *destructive* components of the beat cycles (i.e. the beat minima, as previously suggested by *Field et al., 2019*). Because the phase of interacting EODs is affected by both position and movement, phase shifts induced by chirps could be also aimed at recovering blind-spots forming on the electric images as a consequence of particular body positions, obstacles or even by motion (we show in fact that chirps can improve the EI contrast). Although chirp production has been previously reported to be phase-invariant (*Metzen and Chacron, 2017*), invariance here means that chirps are not produced at a specific EOD phase. Yet, this does not exclude the possibility that chirps could be used to briefly shift the EOD phase in order to avoid disruptive interferences caused by phase opposition (at the level of p-units). Therefore, this *chirps-as-saccades* idea could be still a valid hypothesis if one considers the effects of beat phase adjustment only at specific body locations (such as the head, which can be considered the 'electrosensory fovea'; *Carr et al., 1982*; *Castelló et al., 2000*). In this view, chirping could be seen as an attempt to adaptively modify the incoming sensory input and maintain optimal contrast levels or a coherent sensory flow, as sniffing does for olfactory perception (*Hahn et al., 1994*), whisking and finger movement for tactile sensing (*Krammer et al., 2020*) and saccades for vision (*Najemnik and Geisler, 2005*). Future studies focused on the details of the EOD geometries and fish spatial coordinates during chirping will shed more light onto these matters.

In theory, chirps could also be used to improve electrolocation of objects as well (as opposed to the processing of the beat). Compared to other electric fish using pulse-type EODs, the frequency content of wave-type EODs is narrowly focused on limited components. The larger signal bandwidth (i.e. frequency content) of pulse EODs affords better object discrimination, when compared to wave-type EODs. However, given their lower production rates compared to the frequency of wave-type EOD cycles, EOD pulses grant a lower sampling rate and thus a poorer temporal resolution (*Crampton, 2019*; *Bastian, 1976*; *Watson and Bastian, 1979*). To compensate for this, EOD pulses are emitted at higher frequency during electrolocation bouts and locomotion in general (*Jun et al., 2014*). Conversely, for wave-type fish, chirping could represent a strategy to temporarily compensate for the lower frequency resolution while still being able to resolve EOD perturbations with a good temporal definition. Conducting combined video and playback experiments with a more specific focus on target-reaching behaviors could potentially elucidate whether this is the case. In addition, testing beat detection capabilities at different frequencies (or DFs) would be another important experimental approach to evaluate the probing hypothesis.

If this idea finds further support, the question arises as to why not all gymnotiforms electrolocate using the same strategy. It may be that a trade-off exists between space- and time-resolution in the evolution and maintenance of electrosensory systems (*Crampton, 2019*). While broadband pulse signals may be useful to capture highly complex environments rich in foliage, roots and other structures common in the more superficial habitats in which pulse-type fish live, wave-type EODs may be a better choice in the relatively simpler river-bed environments in which many wave-type fish live (e.g. the benthic zone of deep river channels; *Crampton, 2019*). In this case, achieving a good spatial resolution is critical during social encounters, especially considering the limited utility of visual cues in these low light conditions. In such habitats, social encounters may be less 'abrupt', but spatially less 'conspicuous' or blurred (as a 3D electric field may be). In such a scenario, chirps could serve as

a means to supplement the spatial information acquired via the beat, accentuating these cues during periods of reduced resolution.

## Conclusions

While our results do not completely rule out the possibility of chirps serving a role in social communication, the high stereotypy and auto-correlation of chirp time-series, along with the absence of any meaningful social correlate of chirping, suggests this scenario to be unlikely. Given the clear distance threshold at which chirps are produced, the correlation with locomotion and the increase of chirping in cluttered environments, chirp production mirrors the emission of other self-referenced signals used to gather spatial information obtained from a nearby source (such as echolocation calls; *Siemers et al., 2009*). Could chirps be useful for both electrocommunication and electrolocation functions? Or more simply, could chirping be ascribed simultaneously to different functional tasks?

Signals with such dual function (i.e. echolocation and behavioral referencing) may be a rare find in nature (eavesdropped signals can become cues for localization purposes but are not intentionally produced with that aim), whereas it is more likely that signals directly affecting both senders and receivers do so by delivering the same type of information (i.e. spatial, in this case). In this scenario, probing cues could function simultaneously as proximity signals to signal presence, deter approaches, or coordinate behaviors like spawning, if properly timed (*Henninger et al., 2018*).

Conversely, if chirps would be referenced to specific behavioral states – as posed by the *communication-hypothesis* – the variable rate of chirp production across individuals and the infrequent production of chirp types other than type 2 chirps would render such a communication channel unreliable, semantically limited, and potentially incompatible with previously attributed functions of chirps.

In echolocating species, the effectiveness of active sensing has been demonstrated to rely significantly on environmental complexity and spatial features (*Fenton and Bell, 1981*; *Siemers et al., 2009*; *Fouda et al., 2018*). Considering chirps as probing cues, knifefish might adaptively use different types of chirps based on the features of the foreign EOD source (frequency, position, location, orientation, distance, etc.) and the structure of the habitat. We emphasize the role of chirps as homeoactive signals – as opposed to alloactive – to underline the fact that chirps represent active modulations of signal frequency possibly aimed at optimizing the otherwise passive responses to the beat, occurring continuously and unintentionally (*Zweifel and Hartmann, 2020*).

In summary, although our behavioral experiments may not offer a comprehensive causal understanding of chirps and behavior, they do shed light on the potential and, in part, hitherto unexplored roles of chirping. The aim is to stir the pot and initiate a discussion on possible alternative functions of chirps beyond their presumed communication role.

## Materials and methods

### Animals

A total of 234 *Apteronotus leptorhynchus* of both sexes – age ranging between 2 and 3 years – were obtained from tropical fish importers and housed in individual 80x35 x 40 cm aquariums under a 12/12 light cycle. Fish were allocated to the different experiments as follows: 130 for the assessment of context-dependent effects on chirping, 16 fish were used for the playback experiments, 14 fish were used for the playback experiments involving EOD frequency ramps, 24 fish were used for the chirp-behavior correlative analysis, 30 fish were used in the novel environment exploration assay, 12 fish were used to assess the environment effects on chirping and 8 to assess the impact of clutter on the fish responses to playback EODs. In all groups both sexes were equally represented. Throughout the experiments, the water was continuously filtered, water conductivity was maintained between 150–300 µS cm$^{-1}$, temperature between 23–26°C and pH between 6–8. The hometank environment consisted of PVC shelter tubes and plastic plants. The fish were fed three times a week with red mosquito larvae. Prior to each experiment, the fish's EOD frequency (EODf), body length and weight were measured. To identify their sex, the EODf was normalized to a water temperature of 25.0 °C using a Q$_{10}$ of 2 (*Dunlap and Ragazzi, 2015*; *Dunlap et al., 2000*). The limitation of this approach is that females cannot be distinguished from immature males with absolute certainty, since no post-mortem gonadal inspection was carried out. Nevertheless, fish with a normalized EODf higher than

750 Hz are considered males and those with lower frequencies are considered females (*Dunlap et al., 2000*). Although a more accurate way to determine the sex of brown ghosts would be to consider other morphological features such as the shape of the snout (*Hagedorn and Mary, 1986*), the body size, the occurrence of developing eggs, EOD frequency has been extensively used for this purpose (*Hagedorn and Mary, 1986*; *Meyer et al., 1987*; *Bastian et al., 2001*).

## Recordings of fish pairs and playback experiments
### Equipment
All technical equipment used in the playback experiments is listed in the table below.

| Device | Model | Manufacturer |
| --- | --- | --- |
| Amplifier | DPA-2FSL and DPA-2FS | npi electronic GmbH |
| Attenuator | 839 attenuator | KAY elemetrics Corp. |
| Computer | OptiPlex 7050 | Dell Technologies Inc |
| Data acquisition system | NI 6211 USB | National instruments |
| Infrared illuminators | CM-IRP6-940 nm | CMVision |
| Stimulus isolator | ISO-STIM 01 M | npi electronic GmbH |
| Infrared video camera | GS3-U3-41C6NIR-C | FLIR |

### Experimental setup and groups
Fish pairs were recorded in two adjacent chambers of tripartite 80x35 x 40 cm aquariums (*Figures 1, 2 and 5*). Tank compartments were separated by a plastic mesh barrier to allow electrical but no physical interaction. EOD recordings were conducted using 2 graphite electrodes placed on opposite sides of the tank (1 pair per compartment). EOD recordings were assigned to the following categories based on the type of experience fish had with either the test aquarium or the paired subjects: *resident*, *intruder*, *dominant*, *subordinate*, *novel*, *experienced*, *courtship*, *no courtship*, *divided* and *free*. *Resident* fish were housed for 1 week alone in the same setup, before being paired with *intruder* fish. Dominance was assessed by means of shelter-competition tests prior to EOD recordings. These tests consisted of 30 min long trials in which fish pairs were allowed to interact freely and compete for the occupancy of a plastic shelter tube positioned in one of the three compartments of a tripartite aquarium (1 pair per aquarium). The fish spending more time in the shelter tube was considered to be the *dominant* one, the other the *subordinate*. Fish were selected such that one fish was always 3–5 g heavier than the other, to ensure a predictable outcome of the competition and to limit aggressive displays to a minimum (*Jennions and Backwell, 1996*; *Umbers et al., 2013*). In these and other recordings, fish were considered '*experienced*' after at least 1 week of pairing, as opposed to '*novel*' (just paired). This means that residents, intruders, dominant and subordinate fish were all assigned to the category 'novel', whereas the category 'experienced' included only resident and intruder fish, due to their longer pairing period (1 week). In a separate set of recordings, male and female pairs were subject to water conductivity changes to simulate the onset of the breeding season: during the course of a 4-week period water conductivity was lowered from 400 to 100 µS cm⁻¹ (novel fish in high conductivity water = *no breeding*, experienced fish in low conductivity water = *breeding season*). Although, the term 'breeding' here refers to the context and not to the actual behavioral repertoire displayed by the fish, this treatment resulted in 6/8 female fish to show evident signs of developed eggs (no post-mortem exam was used to confirm egg presence in the other two fish) while in other fish cohorts (housed in larger aquariums and with improved shelters) we successfully obtained newly spawned fish. At the end of this period, fish were allowed to swim freely and interact without mesh barriers (experienced fish in low conductivity water = breeding + free). Recordings from such pairs were compared with naïve male-female pairs freely swimming in high conductivity (400 µS cm⁻¹) water (naïve fish in high conductivity water = no breeding +free). Although even in this case fish were paired with their tank partners for longer times, due to the different treatment they had been subject to, they were not included in the 'experienced' category.

**Table 3.** Chirp categorization used for chirp detection.

|  | Frequency excursion < 105 Hz | Frequency excursion > 105 Hz |
|---|---|---|
| Duration > 50 ms | rise | Type 3 chirp |
| Duration < 50 ms | Type 2 chirp | Type 1 chirp |

## Chirp type categorization

Overall, the coding of custom MATLAB scripts (for EOD recording, chirp detection and validation), the preparation and execution of the behavioral experiments and the manual analysis of chirps imposed very significant timing constraints for the execution of the experiments. For this reason, to come up with threshold values usable to categorize chirps over the course of the whole study, we have used a preliminary dataset of 11,342 chirps obtained from our first set of recordings (8 male-male fish pairs). A more complete dataset (N=30,486) was obtained and validated only 2 years later.

The cut-off values used in our study (50ms duration and 105 Hz frequency modulation amplitude) were chosen based on the distribution of our preliminary recorded chirps but also based on reference values previously published by other authors (see *Table 1* for previously published chirp categories and *Table 3* for details on our own categorization).

All chirps recorded (N=67,522,, see *Table 2*) were obtained from different experiments in these proportions: 30,486 from staged fish pair interactions, 15,720 from playback experiments, 3966 from ramp playback experiments, 4672 from freely swimming fish pairs used for chirp-behavior correlation analysis, 7893 from novel environment exploration experiments, 1864 from fish interactions in cluttered environments, 2921 from clutter playback experiments.

## Amplifier and recording settings

EOD recordings (three channels) were amplified through DPA-2FSL and DPA-2FS amplifiers (*npi electronics*) with a gain of 200, low pass filtered at 10 kHz, high-pass filtered at 100 Hz. The acquisition sampling rate was kept at 20 kHz in all recordings.

## Playback experiments

All playback experiments were conducted in 80x35 x 40 cm glass aquariums in a dark room (*Figures 3 and 4*). Water conductivity and temperature were similar to the fish housing conditions and ranged between 200–300 µS cm⁻¹ and 23–26°C, respectively. Water was pre-heated using commercially available heaters (EHEIM 3612 heating rod 50 W). At one end of the tank, a mesh barrier separated the playback electrode from the fish. Electrodes were placed 10 cm apart and 1 cm away from the barrier. This layout ensured that playback stimuli consisted of naturalistic electric fields (*Kelly et al., 2008*). Electrodes were randomly placed at either side of the aquarium, in different experiments, to avoid playback location biases.

Three pairs of recording electrodes were placed in the fish compartment in accordance with an 'IX' layout: one pair was parallel to the short tank side ('I') the other two pairs were oriented diagonally across the compartment ("X"). This layout ensured that the fish EOD was always detectable, regardless of fish movement and body orientation. Prior to commencing the study, the stimulus amplitude was calibrated to have a field intensity of approximately 1 mV/cm measured across the mesh divider at circa 5 cm distance from the playback dipole, and in line with its axis.

Playback sinewave stimuli were designed in MATLAB and their frequency based on the measured fish EOD frequency prior to trial onset. Stimuli were then delivered via a 6211-USB DAQ (National

**Table 4.** Playback chirp parameters.

| mode | duration [ms] | freq. mod. [Hz] | Chirps |
|---|---|---|---|
| 0 | - | - | no chirps |
| 1 | approx. 120 | 500 | 0.1 Hz for 50 s (1 chirp every 10 s=5/trial) |
| 2 | approx. 20 | 120 | 2 Hz for 5 s (n=10), every 10 s (n=10 × 5) |
| 3 | approx. 60 | 30 | 0.1 Hz for 50 s (1 chirp every 10 s=5/trial) |

Instruments) to the aquarium through a stimulus isolator (ISO-STIM 01 M, npi electronics). The stimuli played back were modified sinewaves mimicking conspecific EODs the frequencies of which were calculated as differences (DF) from the fish's own EOD frequency: −240,–160, −80,–40, −20,–10, −5, 0, 5, 10, 20, 40, 80, 160, 240 (Hz). Each of these playback stimuli was delivered in four different modes distinguished by their chirp content (modes, see *Table 4* below): mode0 contained no chirps, mode1 contained type 1 chirps, mode2 with type 2 chirps and mode3 contained small rises (abrupt frequency rises). Each playback trial (50 s with a 5 s time of fade-in and fade-out) had a 180 s inter-trial interval. Each playback session consisted in a randomized sequence of 15x4 playback trials (1 min+3 min each) which lasted for 4 hr.

### Video tracking

During playback experiments (*Figure 3*) but also freely swimming interactions (*Figure 6*) and novel environmental exploration assays (*Figure 8*), freely swimming fish were recorded at 40 FPS using an infrared USB camera (Grasshopper3, FLIR, model: GS3-U3-41C6NIR-C). The camera was set to acquire a frame every 500 samples of the EOD recording (1 s=20,000 samples) through a TTL channel of the digital acquisition device (NI USB 6211 National Instruments). Swimming trajectories were extracted from the videos using the software BioTracker after converting the files from H264 to AVI (https://www.igb-berlin.de/en/biotracker). Playback experiments began around 1 p.m., after the onset of the animals lights OFF phase (12 AM lights ON, 12 PM lights OFF). Test subjects were acclimated for 30 min to the test aquarium, before trial onset.

### Ethogram used to analyze freely swimming behavioral interactions

During freely swimming interactions, behaviors were annotated following previously described reports (*Triefenbach and Zakon, 2008*). A total of 12 fish pairs were first habituated for 30 min to the testing aquarium (30x80 cm) in presence of a mesh divider. Right after, the divider was removed and fish were allowed to interact freely for another 30 min. Both their EODs and behaviors were recorded in both sessions. During the habituation session, most fish remained in proximity to the barrier, parallel to it, swimming back and forth or stationary. Some fish remained stationary in a corner of their tank compartment, others swam more often exploring their surroundings. In the freely swimming session, fish often continued their exploration rounds before interacting more closely. This often implied a repeated circling of the whole tank contour. Interactions often started with backward swimming (a behavior we referred to as 'knife'), often resolving in a reciprocal backward tail-to-tail approach ('tail1'). In some cases, one of the two fish would flicker the tail in rapid movements ('tail waving', or 'tail2'). From this positioning, fish would often slide onto each other to begin a more direct physical interaction ('wrestle'). This would often imply locomotion ('loc') towards one end of the tank. When reaching a corner fish would either disengage and part ways or start an aggressive interaction, visible through either head butts (one fish hitting with the snout the flank of the other, 'head') or – somehow more frequently – though bites directed to the head of the other fish ('atk'). Less commonly, the attacking fish would quiver their jaws ('jaw') prior to either an attack (directed to the head or the flank) or a chasing attempt ('ch'). The number and extent of such encounters varied considerably across fish pairs: in some cases, it lasted for the whole duration of the recording, while in others fish were just resting at opposite sides of the tank.

### Boundary element model and electric images

The results shown in *Figure 7* (electric images) and *Appendix 1—figure 13* (electric field) were calculated using the boundary element method (BEM; *Pedraja et al., 2014*; *Hofmann et al., 2017*; *Rother, 2003*). This model comprises two main components: a geometric reconstruction of the fish's body and a calculation of the transcutaneous field by solving the Poisson equation for the fish boundary.

Briefly, the BEM method determines the boundary electrical distributions by solving a linear system of M x N equations, where M represents poles and N represents nodes. The unknown variables in this system are the trans-epithelial current density and potential at each node. These variables are computed for each node and linearly interpolated for the triangles defined by the nodes, which form the geometry of the fish and objects. Fish geometry as well as internal and skin conductivities were modeled using realistic information.

Furthermore, the BEM method allows for the concatenation of multiple instants, enabling the rendering of an entire sequence of electric fields and resulting electric images incorporating realistic electric organ discharge (EOD) characteristics such as waveform, duration, and frequencies (see *Pedraja et al., 2014*).

Here, electric images were computed for each fish position and chirp scenario by simulating various phases (the simulations started with eight equally spaced phase relationships between the sender and receiver EODs). The integral of the voltage change over time was calculated throughout the entire duration of the chirp (analysis window = 500ms), ensuring consistent timing for both chirp and non-chirp conditions across 835 body nodes.

The "Δ chirp-beat" shown for the chirp sender and the receiver in *Figure 7E* represents the cumulative change in voltage across the entire body surface during a chirp, compared to the same time window without a chirp.

## Beat decay estimates

The beat dissipation range in our experimental conditions was estimated by recording the interaction of two EOD mimics (a static reference electrode pair and another identical electrode pair, placed parallel to it, as a moving source) at increasing distances and in the same water conditions as in our recordings (conductivity 200 µS/cm and temperature 25 °C; *Appendix 1—figure 13*). Both the reference and the moving EOD mimic were scaled to natural fish signals (1 mV/cm, measured at 2–3 cm from the playback dipoles). Recordings were made placing recording electrodes at opposite poles of the reference electrodes while shifting the moving source with 5 cm steps up to 60 cm away. The EOD field intensity generated by both reference and moving electrodes was recorded for 5 s at each distance step, always at the extremities of the reference pair.

## Environmental manipulations

Fish pairs were exposed to changing environmental conditions in the experiments illustrated in *Figure 9*. Recordings in lights ON, OFF and cluttered conditions were carried out in 80x35 × 40 cm aquariums divided in 2 compartments by a plastic mesh divider. During *lights-ON*, even illumination of both tank compartments was provided via a Leica LED illuminator located on a shelf above the aquarium (Leica CLS 150 XE Microscope Cold Light Source, 150 W). During *lights-OFF* and *cluttered* conditions, a drape obtained from a blackout curtain was used to fully cover the tank (room lights were switched OFF and daylight lowered by curtains). Cluttered conditions were exactly the same as *lights-OFF* with the addition of floating and submerged plastic shelter tubes and plants heterogeneously arranged in order to fill up each compartment.

## Data analysis

### Chirp detection

Chirps were detected in a two-step process consisting of a first automatic detection followed by a manual validation. The first step implied a measurement of the power density in the FFT of the recorded signal (FFT parameters: window size = 2^12; overlap = 90%) within the range set by the fundamental frequency and the first harmonic component (EODf - 5 Hz and 2 x EOD - 100 Hz). Chirps were detected as peaks in the power density. The manual validation consisted of a survey of the detected chirps to eliminate false positives. False negatives were estimated at a rate lower than 10%. Although the two electrode pairs used in each aquarium would detect EODs from both fish, chirps were properly assigned to the sender based on signal intensity and the baseline EOD frequency of the two fish, when possible (EODs generated by a fish will have higher intensity if recorded by electrodes placed in the same compartment). Notably, other authors reported the occurrence of more than two types of large chirps (namely: type 3-6), we included all those in one single group based on the distribution of our data and considering the absence of any obvious clustering for large chirps (see *Figure 1*).

### Factor analysis mixed data - I

Factor analysis was conducted using the R package FactoMiner (*Pagès, 2004*; *Josse and Husson, 2008*).

Factor analysis of mixed data (FAMD, *Figure 2*) is equivalent to a principal component analysis adapted to analyze mixed datasets, containing both quantitative and qualitative variables (*Pagès, 2004*) and explore their associations. The FAMD algorithm can be seen as a hybrid of principal component analysis (PCA) and multiple correspondence analysis (MCA): it works like PCA with quantitative variables and MCA for qualitative ones. Quantitative and qualitative variables are normalized in order to balance the influence of each set.

A total of 30,486 sampled chirps were used. For each chirp, the following qualitative and quantitative variables were considered: frequency_modulation (peak of EOD instant frequency during chirp), chirp duration (instant frequency peak duration), EOD frequency of the sender (EODs), EOD frequency of the receiver (EODr), EOD_amplitude (amplitude of the EOD sinewave during chirp), power (chirp triggered EOD intensity), water temperature, water conductivity, chirp timestamp, weight and length of the sender fish, weight and length of the receiver fish, sender and receiver sex, status of the sender and receiver fish (experience with the recording aquarium), fish ID (to account for interindividual variability), chirp type (categorized as above), state (based on the experience with the other fish of the pair).

A first model (three dimensions, model#1, dim1 17.46%, dim2 15.76%, dim3 10.67%) was made considering all variables listed above. Subsequent models were calculated while progressively reducing the number of variables (down to 3, model#11, dim1 23.17%, dim2 20.4%, dim3 14.35%). At each iteration variables with eigenvalues lower than the mean average contribution were eliminated.

## Factor analysis mixed data - II

To assess the correlations among swimming related variables and chirps, a total of 7893 chirps obtained from a separate experiment (novel environment exploration, *Figure 8*) were used. For each chirp, the following qualitative and quantitative variables were considered: the percentage of time spent near the tank wall ('wall'), the percentage of time spent in the tank open space ('open'), percentage of time spent in the shelter area ('shelter'), the percentage of time spent within a rectangular area 10 cm around the conspecific location ('social'), the percentage of time spent investigating the novel object ('object'), the time at which chirps were produced ('timestamp'), the chirp type, the fish ID, the EOD frequency of the freely swimming fish, the EOD frequency of the caged fish, the sex of the chirp sender, the sex of the chirp receiver, the DF (i.e. EOD frequency difference between sender and receiver), the distance between sender and receiver ('distance'), the average speed of the sender ('avg speed'), the angle between sender and receiver (referred to the a horizontal axis delineated by the tube, 'angle'), the interference estimate weighed on the distance (i.e. calculated on a beat resulting from EODs attenuated as a function of distance, 'actual interference') and the maximum beat interference possible for a given chirp (i.e. without distance attenuation, 'interference', *Appendix 1—figure 12*).

## Chirp transitions and time-series cross-correlation

The number of chirp transitions present in each recording (dataset used for *Figures 1, 2 and 5*) was measured by searching in a string array containing the four chirp types per fish pair, all their possible pairwise permutations (i.e. all possible permutations of 4+4 = 8 elements are: 1–1, 1–2, 1–3 … 7–6, 7–7, 7–8; considering the following legend 1=fish1 type 1, 2=fish 1 type 2, 3=fish1 type 3 … 6=fish2 type 2, 7=fish2 type 3 and 8=fish2 rise). The number of each chirp-type transition was then used to create a 4x4 frequency matrix. Chirp transition diagrams were obtained by calculating the median of such frequency matrices calculated for different fish pairs. Chirp time series were extracted from each recording and binned (50ms) to evaluate the cross-correlation index within a +–2 s lag window.

## Chirp interference

Chirp interference was calculated as the cumulative duration of outlier beat inter-peak intervals (IPI) after calculating the beat envelope for each chirp/DF combination (*Appendix 1—figure 9*). Gaussian chirps were generated on top of a constant frequency carrier of 862 Hz (arbitrarily chosen) and the beat resulting from a paired sinewave with a frequency within ±300 Hz of 862 Hz. The frequency modulation and duration of gaussian chirps were determined by setting the sigma and the amplitude of the peak and by measuring the actual peak size of the instant frequency ('medfreq' MATLAB

function). This was obtained from the FFT analysis of the chirping signal (window size 2^12, 90% overlap, sampling frequency 20 kHz). For each chirp, beat peaks (i.e. the peaks of the amplitude modulation, AM) were detected searching for local maxima in the signal first-order envelope. Adjacent beat peaks were used to calculate beat cycle durations for the whole segment. The different impact of different types of chirps on the beat would result in different populations of beat cycles (depending on the beat frequency, the duration and FM of the chirp). Outliers in the beat cycle durations resulting from each chirp were used as a measure of how a given chirp would perturbate the otherwise regular beat cycles. Beat cycle outliers were detected using a MATLAB built-in function – 'isoutlier' – with the 'quartile' setting to include peak durations on both tails of the peak distribution (below 25% and above the 75% quartiles). The interference value for each chirp was obtained by dividing the cumulative outlier beat cycle duration (i.e. the total duration of beat cycles significantly affected by the chirp) by the overall duration of the beat peaks. The duration of each beat cycle corresponded to the difference in time between consecutive beat peaks. In this way, the effect of a chirp could be weighted on the number of peaks (i.e. the beat frequency). The beat interference is calculated within a fixed time window (700ms, but see *Appendix 1—figure 10* for wider windows) which corresponds to the median inter-chirp interval (ICI; median = 0.6987) calculated for 30486 chirps (130 chirp time-series) and is centered around the chirp (chirp peak is always at 350ms). This parameter is used to provide an estimate of the effect of a chirp in its immediate temporal surround.

### Text-mining analysis of chirp-behavior sequences

A text-mining approach was used to identify correlations between behaviors and chirp types in text strings containing both (pooled data from 12 fish pairs of mixed sexes) using KH Coder (*Higuchi, 2016*). KH Coder is open-source software designed for computer-assisted qualitative data analysis, with a focus on quantitative content analysis, text mining, and applications in computational linguistics. After creating a word dataset including all the occurrences of chirps and behaviors in string form (words) conserving their temporal order, we have used KH Coder to perform a co-occurrence analysis and a multidimensional scaling. The first aims at analyzing text identifying potential relationships between words. By definition, co-occurrence networks are represented as interconnected networks based on the paired presence of words within a specified unit of text. The second, multidimensional scaling (MDS), is a means of visualizing the level of similarity of individual words based on their appearance pattern within a dataset. MDS is used to translate information about the pairwise 'distances' among a set of words into a configuration of points mapped into an abstract cartesian space.

### Statistics on chirp counts

The numbers of chirps produced during the experiments assessing the effect of environmental factors on chirp rate production were compared using the Friedman's test (non-parametric ANOVA for repeated measures) as data were not following a Gaussian distribution.

## Acknowledgements

We would like to thank Len Maler, Jan Benda and Jan Grewe for the insightful discussions during the course of this study. We are grateful to Len Maler, Ben Arthur and Jörg Henninger for their constructive comments on the first manuscript draft. Additional thanks to Jörg Henninger for the useful MATLAB "beginner tips". We are especially grateful to Ina Seuffert, Sebastian Kraft, Helen von Drenkmann and Domi Bekaan for their invaluable help with the fish care. We also wish to thank the three anonymous reviewers for their constructive comments to the submitted version of this manuscript. Research funding was provided by the Cluster of Excellence NeuroCure and by Humboldt Universität zu Berlin to R.K. The article processing charge was funded by the German Research Foundation (Deutsche Forschungsgemeinschaft, DFG) - 491192747 and the Open Access Publication Fund of Humboldt Universität zu Berlin.

## Additional information

### Funding

| Funder | Grant reference number | Author |
|---|---|---|
| Deutsche Forschungsgemeinschaft | 491192747 | Rüdiger Krahe |

The funders had no role in study design, data collection and interpretation, or the decision to submit the work for publication.

### Author contributions

Livio Oboti, Conceptualization, Resources, Data curation, Software, Formal analysis, Supervision, Investigation, Visualization, Methodology, Writing – original draft, Writing – review and editing; Federico Pedraja, Marie Ritter, Marlena Lohse, Lennart Klette, Data curation, Formal analysis; Rüdiger Krahe, Supervision, Funding acquisition, Writing – review and editing

### Author ORCIDs

Livio Oboti (ID) https://orcid.org/0000-0001-7197-568X

Reviewer #1 (Public Review): https://doi.org/10.7554/eLife.88287.5.sa1
Reviewer #2 (Public Review): https://doi.org/10.7554/eLife.88287.5.sa2
Reviewer #3 (Public Review): https://doi.org/10.7554/eLife.88287.5.sa3
Author response https://doi.org/10.7554/eLife.88287.5.sa4

## Additional files

### Supplementary files

MDAR checklist

### Data availability

All information required to replicate these experiments has been provided in the Materials and methods section of the manuscript. The data used to prepare the figures in this manuscript and the raw voltage data related to our first set of recordings (compressed in WAV format) can be accessed on Open Science Framework. Interested parties who would need to access the original recordings could do so by contacting the corresponding authors (see below). A link to the data required will be made available. Data are in binary format and include multichannel recordings. The main code used to record, detect and analyze chirps can be found on GitHub, copy archived at *loboti, 2024*. For any request, you are welcome to reach out to us: livio.oboti@gmail.com, rüdiger.krahe@hu-berlin.de.

The following dataset was generated:

| Author(s) | Year | Dataset title | Dataset URL | Database and Identifier |
|---|---|---|---|---|
| Oboti L, Ritter M, Lohse M, Klette L, Krahe R | 2024 | Livio Oboti's Quick Files | https://doi.org/10.17605/OSF.IO/ZASXQ | Open Science Framework, 10.17605/OSF.IO/ZASXQ |

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

# Appendix 1

## Detecting beats at a distance

In weakly electric fish, the electric fields generated by electric organs are typically in the range of a few mV, depending on the species. Brown ghosts generate EODs with an amplitude of roughly 5 mV (*Rasnow and Bower, 1996*) as measured at the head and tail extremities of the fish body. Considering that the decay of dipole moments is regulated by the following law:

$$V = \frac{q}{\varepsilon_0} \cdot \frac{s \cdot cos\theta}{4\pi d^2} \tag{1}$$

Where *V* is the voltage measured at a distance *d* (m), *q* is the dipole charge (in Coulombs), *epsilon* is the vacuum dielectric constant, *s* is the distance between the dipole charges (m), and *theta* is the angle of the vector joining the point at distance d and the dipole midpoint (*Knudsen, 1975*). While this equation is useful to describe the static dependency of field intensity on distance for any measuring point in space, electric fields can also be represented as the vector sum of electric gradients. The formula describing field gradients can be obtained from the formula above by resolving it for each charge in the system with reference to a given point in space moving through it. The field is instantaneously generated by the 2 dipole charges, and so can be described by the formula (*Knudsen, 1975*):

$$E \approx \frac{q}{d^2} \cdot \left[ \left(1 - \frac{2s}{d}\right) - \left(1 - \frac{2s}{d}\right) \right] = \frac{-4 \cdot q \cdot s}{d^3} \tag{2}$$

*Equation 2* clearly shows that within an electric field, the voltage drops with the cube of distance. At double the distance, the field amplitude drop is 8-fold (see *Benda, 2020* for a recent and exhaustive description of the physical properties of weakly electric fields). According to these considerations, the range at which brown ghosts can realistically detect other electric fields depends on whether or not active electrolocation is used. In fact, while the dipole moment generated by an EOD alone dissipates according to *Equation 2*, the amplitude modulation (AM) resulting from the interaction of 2 or more EODs depends on the decay constants of all composing signals. Now, it is known that brown ghosts are not capable of detecting single EODs and reading their absolute voltage values, rather they sense the AM generated by the interacting EODs (i.e. the *beat*). From the amplitude of these AMs (i.e. the *beat contrast*), brown ghosts can infer the relative distance, the size and even the rate of motion of an approaching conspecific (*Assad et al., 1999*; *Babineau et al., 2006*; *Kelly et al., 2008*). The beat contrast can be easily calculated as the ratio of the amplitude of the detected signal and the EOD amplitude of the receiver fish ($A_1/A_0$x100). Behavioral experiments have shown that responses to beats can be elicited in different knifefish species at electric field gradients as low as a few µV/cm (0.6 µV/cm in *Eigenmannia virescens*, *Kaunzinger and Kramer, 1995*; 0.2 µV/cm *Sternopygus macrurus*, *Fleishman et al., 1992* and *Apteronotus albifrons*, *Knudsen, 1974*). In *Eigenmannia virescens*, behavioral responses to EOD AMs were observed at 0.02–0.03% beat contrast (*Carr et al., 1982*; *Kawasaki, 1997*; *Kawasaki, 1997*), although other authors reported for the same species slightly higher values (0.16% contrast, *Rose and Heiligenberg, 1985*; *Rose and Heiligenberg, 1985*). Although beat detection thresholds in *Eigenmannia virescens* and brown ghosts may slightly differ, estimates of field gradient sensitivity have been estimated to be similarly low in brown ghosts (0.1 µV/cm, *Rasnow and Bower, 1996*). Therefore, we could expect similar beat sensitivity levels also in this species. Based on these considerations, we determined the values of EOD amplitude and beat contrast for a reference fish generating an electric field resulting in an average 5 mV voltage measure around its body and a 1.25 mV/cm field gradient starting at 2 cm distance. The average voltage was estimated from skin measurements conducted in brown ghosts and referenced to a ground electrode placed at "infinite" distance" (*Rasnow and Bower, 1996*). At 2 cm distance and starting from 5 mV, the average voltage drops at 1.25 mV (according to *Equation 1*) so choosing this value as a starting point for field measurements at the same distance represents a slight overestimation (the electric field decays with the cube of distance according to in *Equation 2*). Beat contrast is estimated with reference to the first value and only estimates from further distances are shown.

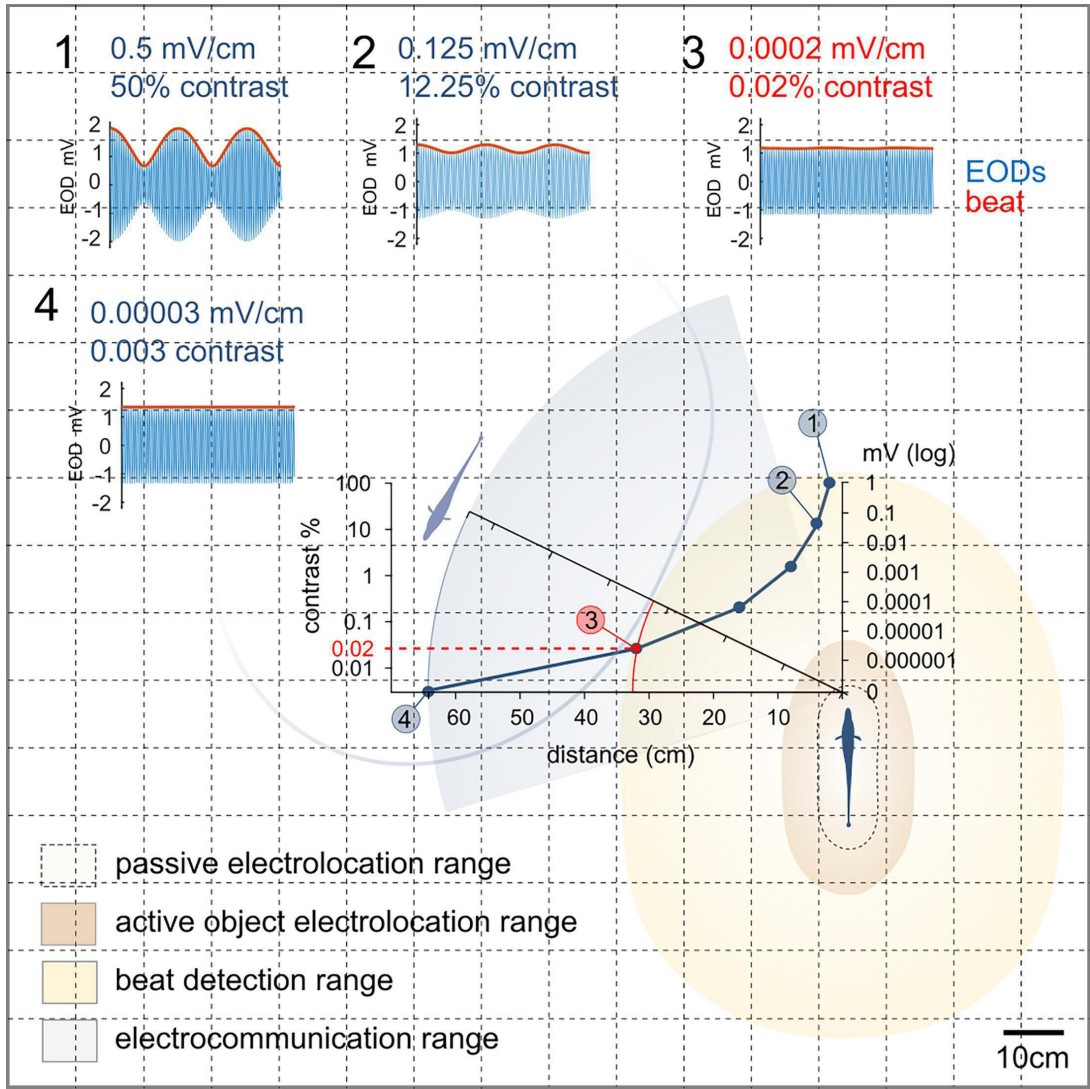

**Appendix 1—figure 1.** Detecting beat at a distance.

In the diagram, electric field ranges are represented by the oval shaped lines drawn around the fish and are based on the voltage values obtained using the equations above. Ranges for passive electrolocation (dotted line) are indicated for reference (*Lissmann and Machin, 1958*; *von der Emde, 1999*; *Knudsen, 1974*). The beat contrast values obtained are plotted as a function of distance. The example illustrates the following situation: a 1.25 mV/cm field is used as reference (on the right) and the beat induced by a neighboring 0.7 mV/cm field at different distances is calculated. The composite signals obtained at representative distances are displayed by the graphs on the top together (blue) with their AM (red). The value in (3) is the lowest possible contrast level at which the moving fish can be detected and corresponds to a limit distance of about 60 cm between the two fish. This estimate is close to the reported range for chirping (32 cm, *Henninger et al., 2018*; *Zupanc et al., 2006*; *Hupé et al., 2008*) and EOD envelope processing (*Fotowat et al., 2013*).

As mentioned, active electrolocation of conspecifics implies the summation of 2 or more EODs and the detection of the resulting beat. Because the maximum amplitude of the resulting composite signals is higher than each component - and because beat contrast is calculated using a different reference depending on the receiver - the threshold for beat detection depends on the decay of the contributing signals and is represented by the minimum AM detectable by each fish (other_ EOD/own_EOD*100). For example, if a "fish 1" producing a 1.25 mV EOD swims near a "fish 2" conspecific having a 0.7 mV EOD, when they are 16 cm apart, the maximum AM produced by the 2 EODs can be roughly EOD1+EOD2 = (1.25+0.0014) mV, which corresponds to a beat contrast

of 0.14%. For fish 2: EOD2+EOD1 = (0.7+0.002) mV, corresponding to a beat contrast of 0,28%. Therefore, at any given distance, different fish will detect beats of different amplitudes. It follows that, if beat detection thresholds can be considered comparable in all brown ghosts, fish with EODs of lower amplitude (usually fish of smaller size such as juveniles or females), will have a wider beat detection range.

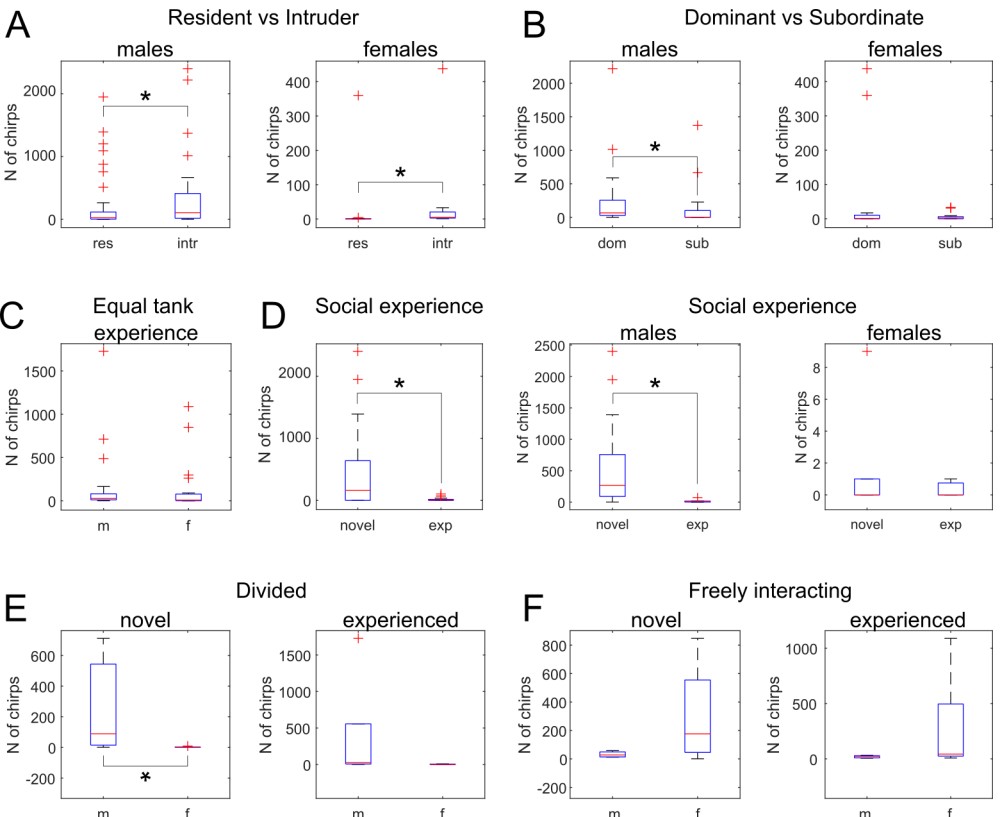

**Appendix 1—figure 2.** Chirping in different social contexts and different tank experience conditions. (**A**) Total chirp counts obtained from resident-intruder assays. Resident fish were housed for 1 week in the test-aquarium while intruders were introduced only at the moment of testing (males, Mann-Whitney U=0.032, females U<0.001). (**B**) Total chirp counts recorded during dominant-subordinate interactions (males, Mann-Whitney U=0.015). (**C**) Total chirp counts obtained from dyadic interactions in which both fish were novel to the test environment. (**D**) Chirp counts evaluated during first time pairing (novel) and after 1 week of pairing (exp, experienced; pooled data, Mann-Whitney U=0.016; males, Mann-Whitney U=0.002). (**E**) Chirp counts obtained from opposite sex pairs at the beginning (novel) and at the end of a 4 week-long water conductivity decrease protocol (used to simulate the reproductive season; novel, Mann-Whitney U=0.034). Fish pairs were interacting only electrically, across a plastic mesh barrier (divided). (**F**) Chirp counts relative to female-male interactions in absence of any mesh barrier (freely interacting). Chirps produced by opposite-sex pairs at the end of the water conductivity changes (experienced) are compared with chirp rates of female-male pairs recorded in absence of prior experience (novel). All chirp counts refer to 1 hour-long recordings except those of dominant-subordinate pairings, which lasted 30 minutes; see Materials and methods for details on the behavioral experiments.

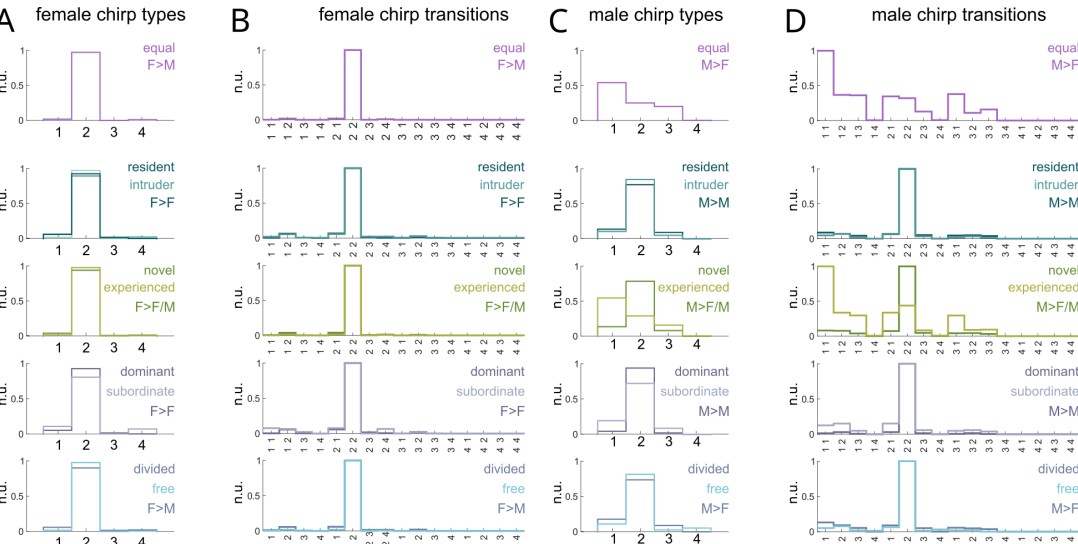

**Appendix 1—figure 3.** Effect of social context and environmental experience on chirping behavior. (**A**) Histograms showing the normalized chirp type distribution relative to female senders and receivers of both sexes (*F*>*F* or *F*>M) in different behavioral contexts. Note the almost identical relative abundance (normalized) of different chirp types. (**B**) Chirp type transitions displayed by female chirpers during different kinds of encounters. In all cases the most common sequence is "type 2-type 2" (the numbers on the X-axis represent the 4 different types, see *Figure 1*). (**C**) Histograms showing the normalized chirp type distribution for male chirpers. The only difference is observed during male-female interactions in conditions of equal tank experience ("equal"). (**D**) Occurrence of chirp transitions in males is similar to those observed in females with the exception of male-female pairs with equal tank experience or longer-term experience with each other. This is likely due to the lower amount of type 2 chirps produced at lower DFs which in turn results in a relatively higher number of larger chirps (chirp type dependency on DF can be better visualized in *Figure 1F* and *Figure 5C and D*).

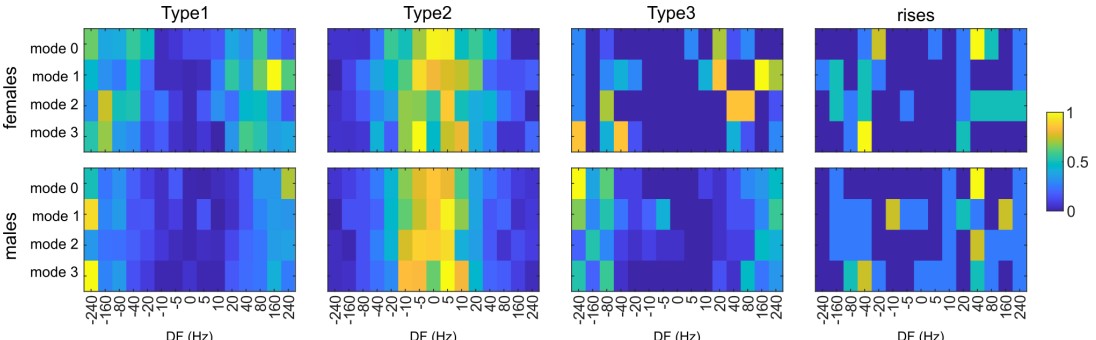

**Appendix 1—figure 4.** Sex comparison of fish responses to playback chirps by DF. Heatmaps of responses to different playback signals sorted by DF (Y-axis) and type of playback chirps (Y-axis, sine = no chirps, type 1, type 2 and rises). The chirps produced by female and male fish are sorted by type and shown in 4 different maps. Chirp counts are normalized on the max value per type. For a description of playback chirps see Materials and methods.

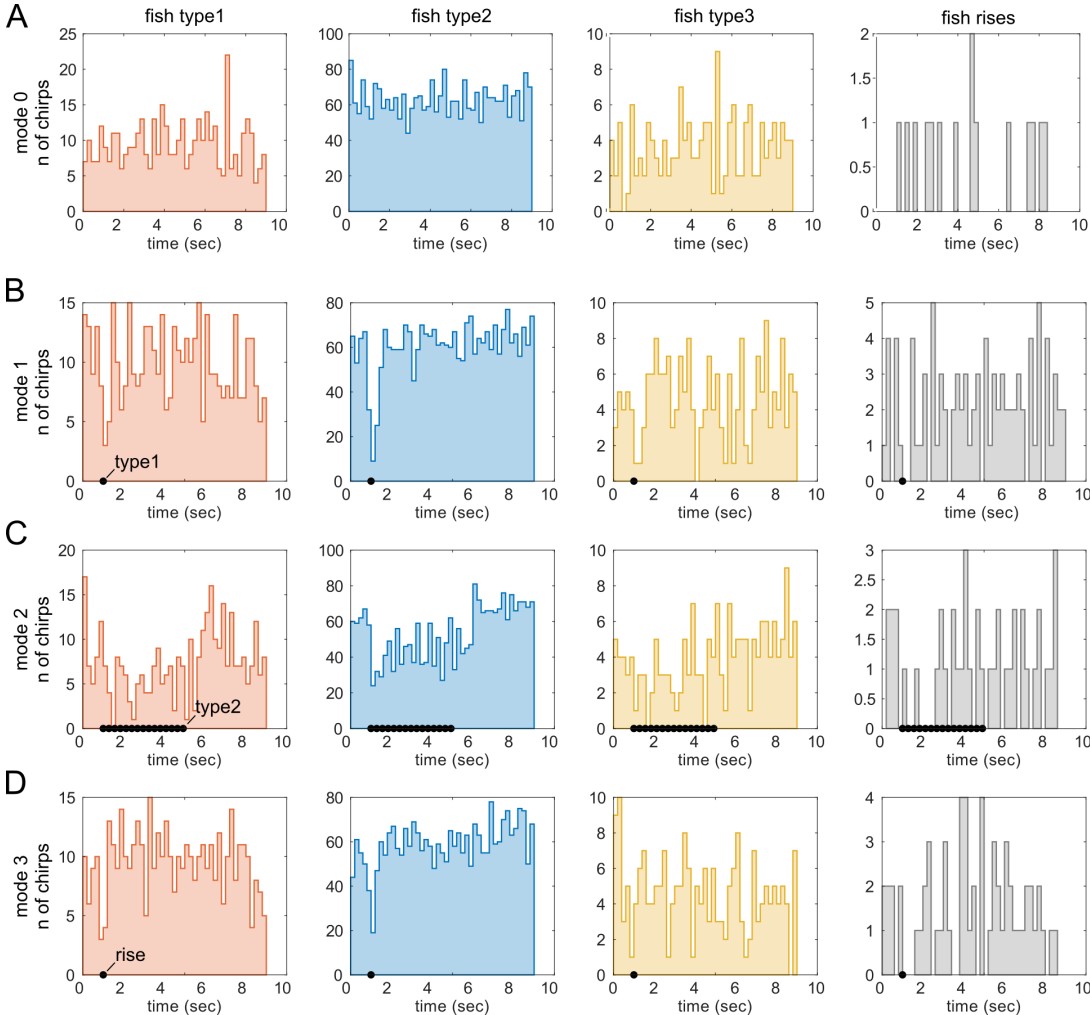

**Appendix 1—figure 5.** Playback chirps temporarily suppress chirping in freely swimming fish. (**A–D**) Peristimulus time-histograms (PSTH) of chirps produced by 16 fish (8 females and 8 males) during playback experiments. Fish responses to different DFs (±240 Hz,±160 Hz,±80 Hz,±40 Hz,±20 Hz,±10 Hz,±5 Hz, 0 Hz) are pooled together. Results for different types of chirps are displayed in different columns while each row is related to chirps produced in response to a given playback mode (mode 0=plain sinewave, mode 1=type 1 chirps at 0.2 Hz, mode 2=type 2 chirps at 3 Hz, mode 3=rises at 0.2 Hz; see Materials and methods for details). The PSTHs show that the main effect of playback chirps (black dots) on the chirps produced by the fish is a brief temporal suppression (accentuated when chirps are repeated in trains, C) but no sign of any significant temporal correlation, except for a transient suppression.

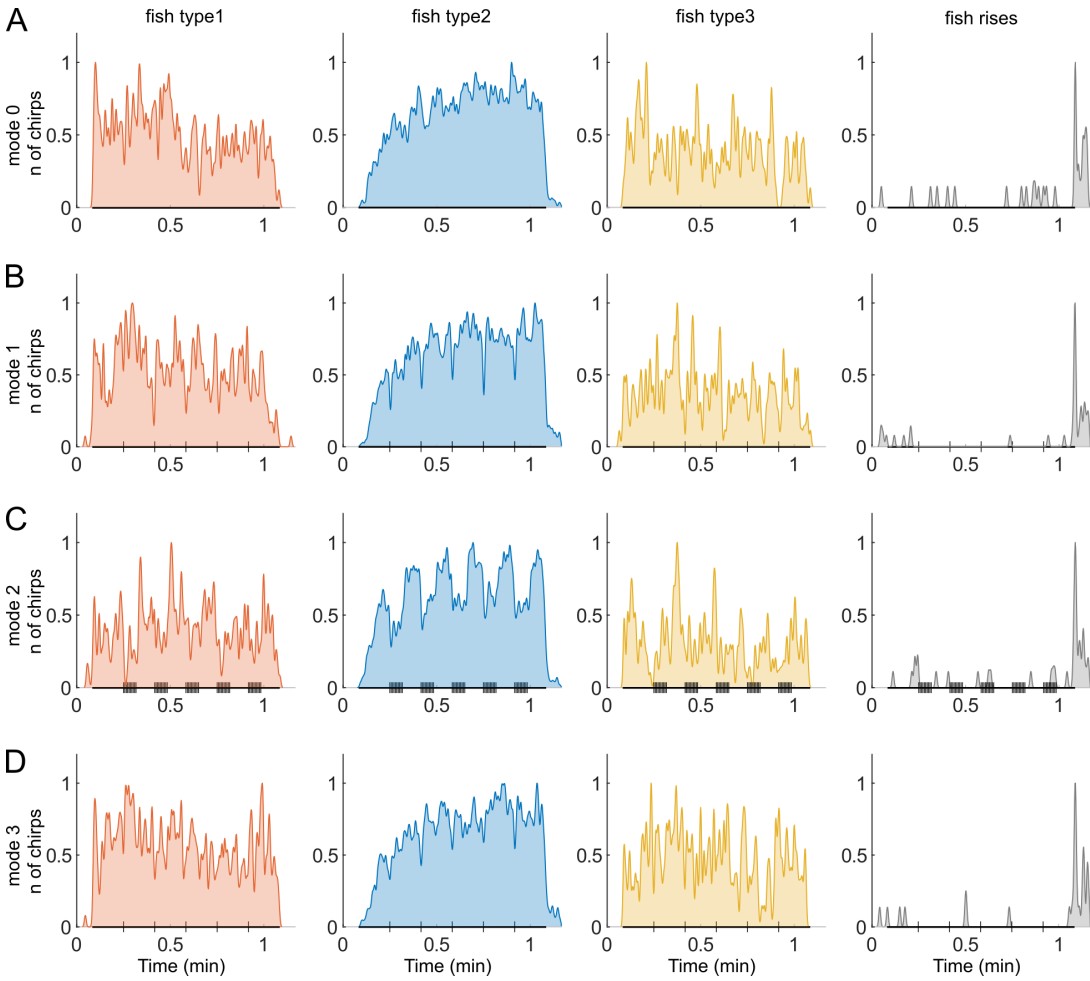

**Appendix 1—figure 6.** Chirp responses to playback EODs containing chirps. (**A**) Average number of chirps of different types (normalized) produced over the course of 1 minute long playback trials (mode 0, sinewave frequency range −240 Hz to +240 Hz). The timing of playback stimuli is represented by the thicker line on the X axis. Vertical ticks on the same axis represent playback chirps. (**B**) Responses to the same set of sine wave EODs to which type 1 chirps were added (mode 1). (**C**) Responses to EODs containing 3 Hz trains of type 2 chirps (mode 2). Note the stronger inhibition of fish chirping exerted by trains of type 2 chirps. (**D**) Responses to EODs containing playback rises (mode 3). See Materials and methods for details on the playback experiments.

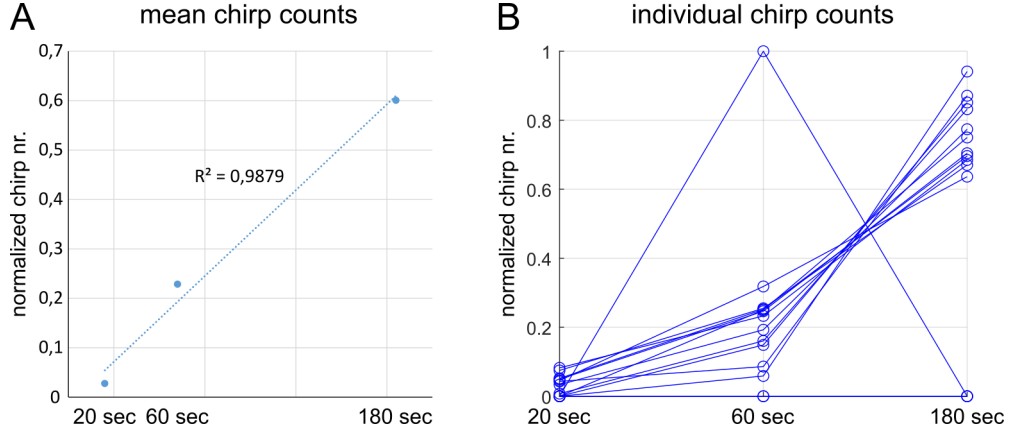

**Appendix 1—figure 7.** Linear correlation of chirp counts and playback duration (playback frequency ramps). (**A**) Correlation coefficient of the mean chirp counts (normalized) recorded as a response to playback frequency
*Appendix 1—figure 7 continued on next page*

*Appendix 1—figure 7 continued*

ramps. (**B**) Line plots related to the individual subjects. The outlier fish emitted 1 chirp during the 60 sec trial and no chirps in the other 2 trials. Fish never producing any chirp were excluded (N=2).

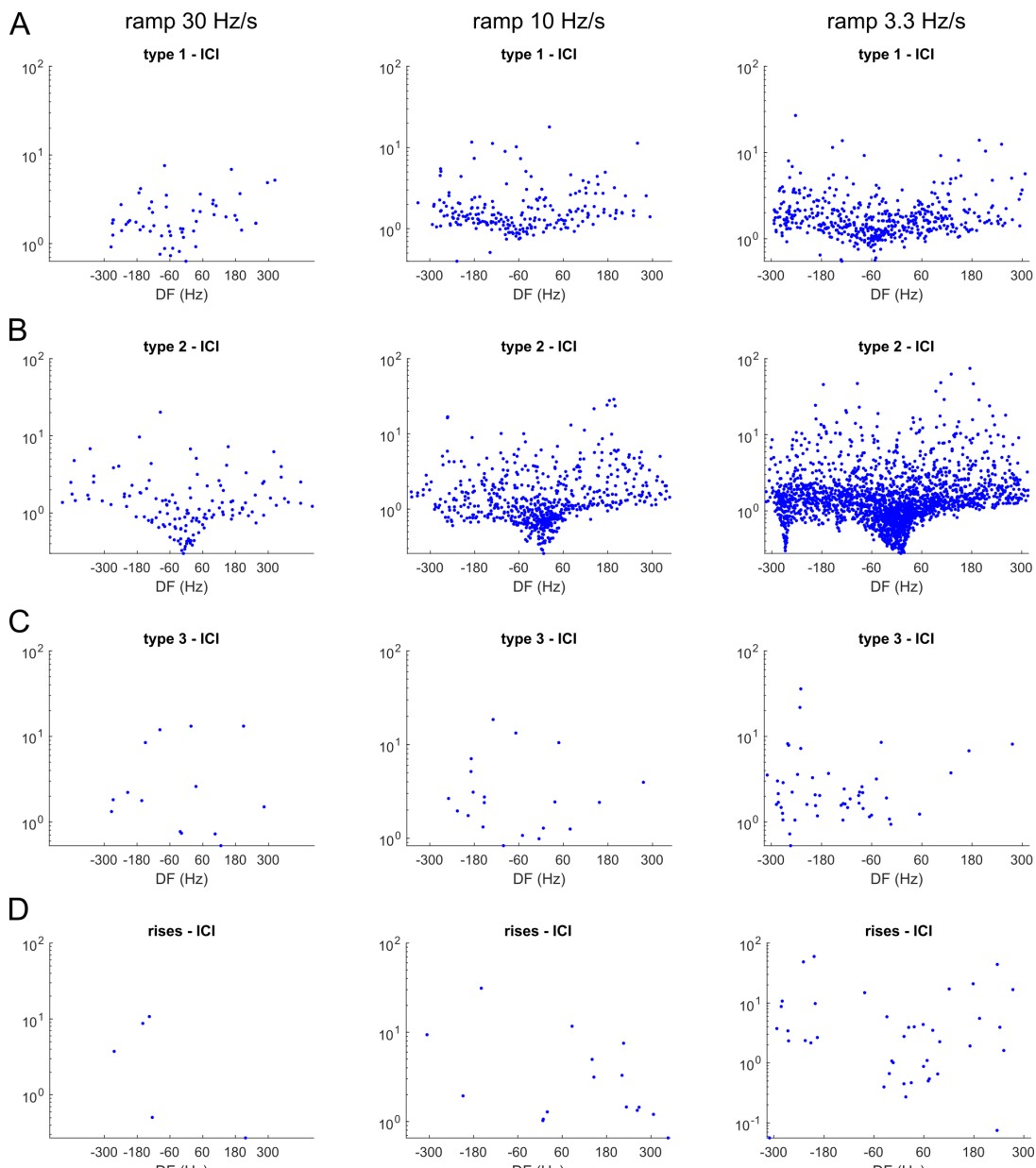

**Appendix 1—figure 8.** Frequency ramp playback experiments, inter-chirp intervals (ICI) of different chirp types. (**A–D**) The inter-chirp latencies following each type of chirp are reported for each of the 3 different frequency ramps (see values on the top). These represent the time passing after a given chirp, before a chirp of any given type would follow. Low ICI values are found more often around the DF = 0 Hz (more markedly for type 2 chirps). Low values are also found at trial onset, due to the often-higher chirp rates observed at the beginning of a playback trial (see type 1 and type 3 chirps in *Appendix 1—figure 6* for instance).

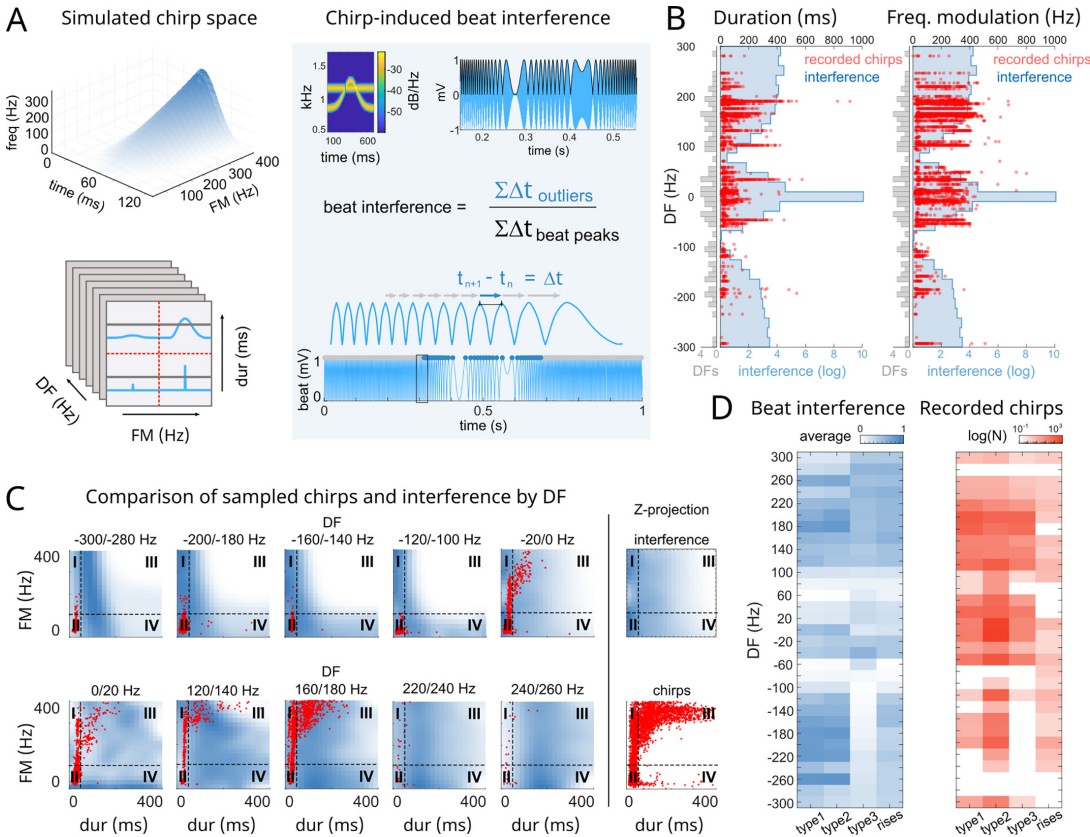

**Appendix 1—figure 9.** Chirp interference with beat regularity. (**A**) Artificial chirps were generated on an 862 Hz EOD baseline covering a 0–400ms duration range and a 0–400 Hz peak amplitude. An additional EOD signal was added to the chirping signal (−300–300 Hz DF range) to simulate a wide range of beat frequencies. The sign of the DF is referred to the baseline reference EOD. The beat interference induced by chirps was calculated as the ratio of the cumulative duration of beat interpeak intervals (IPI) affected by a chirp (i.e. outliers in the IPI population relative to each EOD pair) on the total cumulative beat IPI duration (including outliers) within a 700ms time window (see Materials and methods for details). (**B**) Duration and frequency modulation (FM) histograms of recorded chirps (red, N=30486) sorted by DF and matched to their estimated beat interference (blue). The gray histograms on the Y-axis represent the beat frequencies sampled. (**C**) Normalized heatmaps showing examples of chirp-induced beat interferences (color coded in blue) calculated at different DF values. Real chirps produced at the same DF are overlaid in red. Only overlapping FM and duration ranges are shown. In the plots on the extreme right: beat interference values are summed over all DFs (top) and are shown next to the corresponding chirps (bottom). (**D**) Comparison of average chirp-induced beat interferences sorted by type (left map, blue) and the actual observed chirps (log N, right map in red).

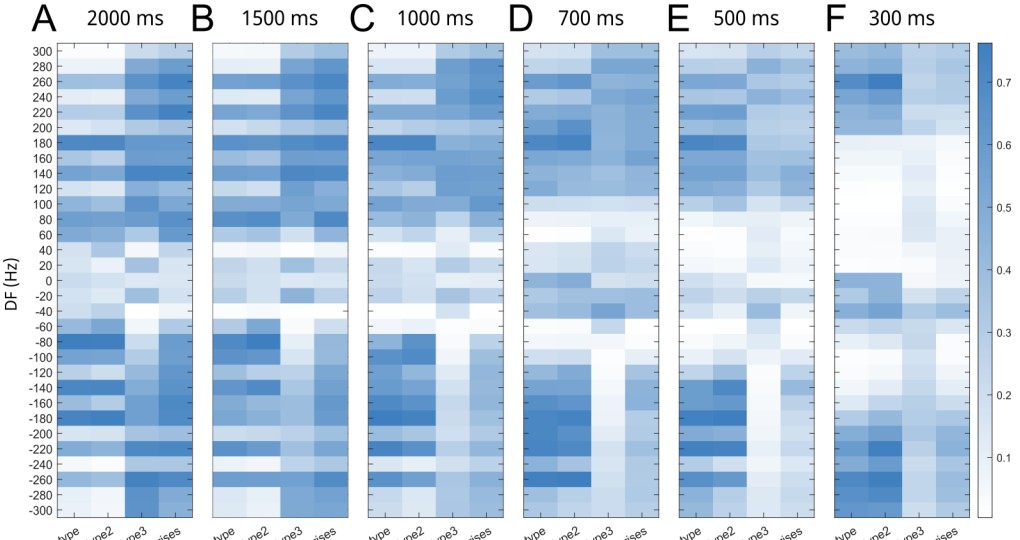

**Appendix 1—figure 10.** Beat interference calculated using different time-windows. The effect of the size of the analysis window on the beat interference estimates for the different types of chirps can be evaluated by comparing the heatmaps above. Since the estimate depends on the number of outliers among the beat cycles included within a fixed time interval, the interference value will decrease/change depending on the number of beat cycles considered (i.e. the size of the analysis window): a small window size will have more outliers for fewer beat cycles but a single chirp will affect a larger number of cycles (**F**). When a larger analysis window is used (**A–C**), the effect at low DFs is diluted as fewer cycles are affected overall. Other factors potentially affecting the interference estimate include: the algorithm used to detect outliers (here, the outliers are detected in the upper and lower quartiles in the distribution of peak durations) and the phase at which the chirp occurs (here the average of 4 phases is used).

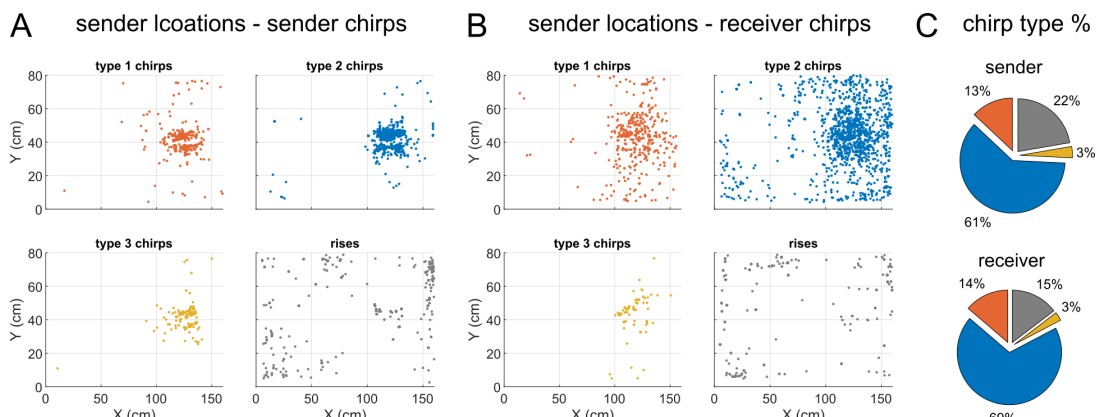

**Appendix 1—figure 11.** Chirp type locations during novel environment explorations. (**A**) Sender fish locations during chirps of different types. Most chirps are produced while brown ghosts are swimming in close proximity to a caged conspecific (see **Figure 8**). This is particularly evident for freely swimming "sender" fish. Chirps produced by caged fish ("receiver") are more widely distributed. Rises are produced when fish are perpendicularly oriented, along the wall (right) or half hidden behind shelters or plastic barriers. (**B**) Sender fish locations during chirps produced by the caged conspecific (receiver). Receiver chirps are produced at similar locations, and in similar percentage (**C**).

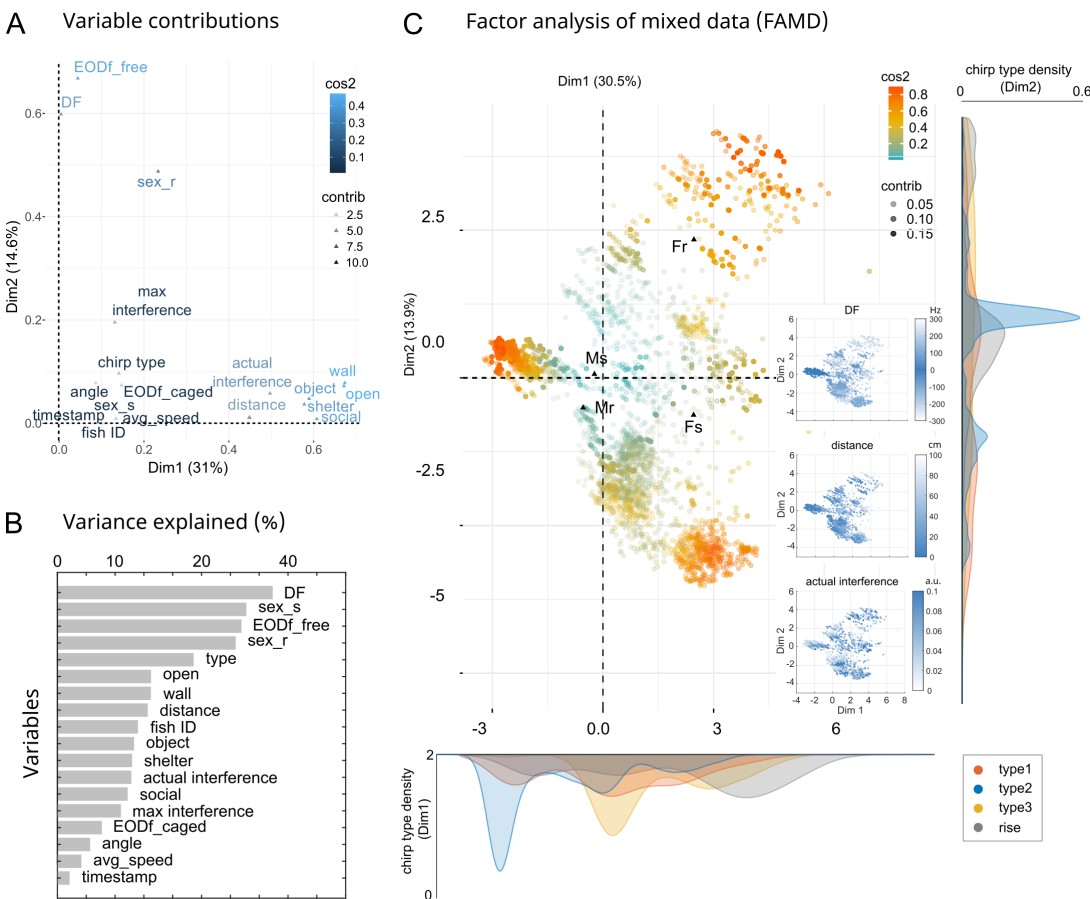

**Appendix 1—figure 12.** Factor analysis of mixed data (FAMD) – novel environment exploration. (**A**) The scatter plot represents the contribution of chirp-related variables to the 2 main components of the transformed space. The contribution of each variable is indicated by both the position and the color of the corresponding marker on the plot. The variables closer to the origin contribute less to the overall sample variance. Variable contribution (contrib) is coded by color intensity and the quality of the representation by color hue (cos2). (**B**) Bar plot showing the total variance (i.e. sum of each variable loading on all the 3 dimensions) explained by each variable in the transformed space. (**C**) Representation of all 7894 chirps in the transformed coordinates. Triangles indicate the coordinates of the qualitative variable centroids (Fs = female sender, Fr = female receiver, Ms = male sender, Mr = male receiver). The contribution of individual chirps is color coded as in B. The clustering is based on both quantitative and qualitative coordinates (fish ID, sex sender, sex receiver, chirp type, average swimming speed, distance and angle between fishes, time spent in the tank ROIs, EOD frequencies of the interacting fish and DF, estimated beat interference and maximum interference possible). The marginal histograms show the kernel density distributions of different chirp types. In the insets, chirps are color-coded according to DF, distance and interference.

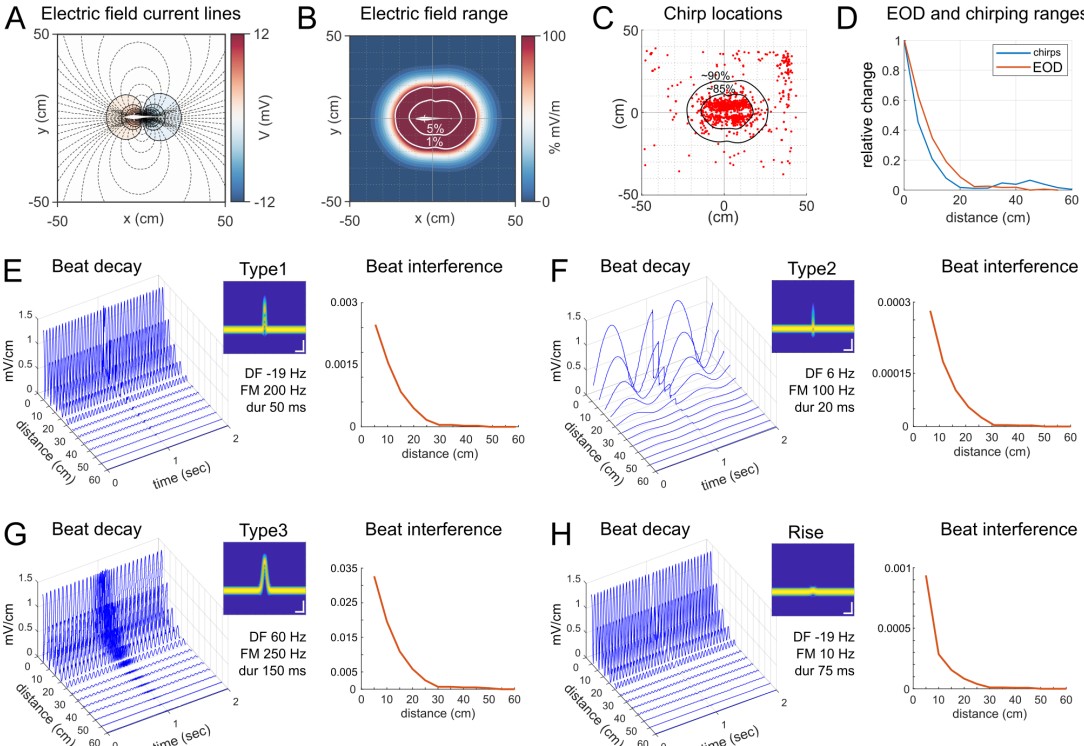

**Appendix 1—figure 13.** Chirps interfere with the beat within the electric field range. (**A**) Electric field generated by a 16 cm 3D dipole modeling an electric fish of the same size using BEM (boundary element methods). Iso-potential lines are shown for the near-field range in different colors, based on field polarity. Current is represented by the dashed lines, perpendicular to them. (**B**) Electric field intensity mapped around the same modeled fish. The level lines represent the 1% and 5% intensity of the electric field generated by the ideal fish. (**C**) Scatter plot of chirp locations. The overlay is centered at the origin and corresponds to 90% and 85% of all chirps produced, respectively. (**D**) Plot showing the intensity range of an EOD mimic calculated at 22–25°C and 200 μS (red) and the distribution range of chirps emitted by real fish (blue), for comparison. (**E–H**) 3D plots showing the ideal beats calculated for different sinewave pairs during different chirp types and plotted over distance. The 2D plots on the side represent the beat interference (calculated using a threshold of 1% of maximum beat amplitude) caused by each chirp type over distance. Scale bars in the spectrograms are 100 ms and 100 Hz.

