## [Editor Report · eLife Assessment]

This study addresses a significant question in sensory ethology and active sensing in particular. It links the production of a specific signal - electrosensory chirps - to various contexts and conditions to propose that chirps may also serve an active sensing role in addition to their more well-known role in communication. The evidence supporting the role for active sensing is strong. In particular, the evidence showing increased chirping in more cluttered environments and the relationship between chirping and movement are **convincing**. The study provides a lot of **valuable** data, and is likely to stimulate follow-up behavioral and physiological studies.

---

## [Referee Report · Reviewer #1 (Public Review)]

The authors investigate the role of chirping in a species of weakly electric fish. They subject the fish to various scenarios and correlate the production of chirps with many different factors. They find major correlations between the background beat signals (continuously present during any social interactions) or some aspects of social and environmental conditions with the propensity to produce different types of chirps. By analyzing more specifically different aspects of these correlations they conclude that chirping patterns are related to navigation purposes and the need to localize the source of the beat signal (i.e. the location of the conspecific).

The study provides a wealth of interesting observations of behavior and much of this data constitutes a useful dataset to document the patterns of social interactions in these fish. Some data, in particular the high propensity to chirp in cluttered environments, raises interesting questions. Their main hypothesis is a useful addition to the debate on the function of these chirps and is worth being considered and explored further.

After the initial reviewers' comments, the authors performed a welcome revision of the way the results are presented. Overall the study has been improved by the revisions.

---

## [Referee Report · Reviewer #2 (Public Review)]

Studying Apteronotus leptorhynchus (the weakly electric brown ghost knifefish), the authors provide evidence that 'chirps' (brief modulations in the frequency and amplitude of the ongoing wave-like electric signal) function in active sensing (specifically homeoactive sensing) rather than communication. Chirping is a behavior that has been well studied, including numerous studies on the sensory coding of chirps and the neural mechanisms for chirp generation. Chirps are largely thought to function in communication behavior, so this alternative function is a very exciting possibility that should have a great impact on the field.

The authors provide convincing evidence that chirps may function in homeoactive sensing. In particular, the evidence showing increased chirping in more cluttered environments and a relationship between chirping and movement are especially strong and suggestive. Their evidence arguing against a role for chirps in communication is not as strong. However, based on an extensive review of the literature, the authors conclude, I think fairly, that the evidence arguing in favor of a communication function is limited and inconclusive. Thus, the real strength of this study is not that it conclusively refutes the communication hypothesis, but that it calls this hypothesis into question while also providing compelling evidence in favor of an alternative function.

In summary, although the evidence against a role for chirps in communication is not as strong as the evidence for a role in active sensing, this study presents very interesting data that is sure to stimulate discussion and follow-up studies. The authors acknowledge that chirps could function as both a communication and homeactive sensing signal, and the language arguing against a communication function is appropriately measured. A given electrical behavior could serve both communication and homeoactive sensing. I suspect this is quite common in electric fish (not just in gymnotiforms such as the species studied here, but also in the distantly related mormyrids), and perhaps in other actively sensing species such as echolocating animals.

---

## [Referee Report · Reviewer #3 (Public Review)]

Summary:

This important paper provides the best-to-date characterization of chirping in weakly electric fish using a large number of variables. These include environment (free vs divided fish, with or without clutter), breeding state, gender, intruder vs resident, social status, locomotion state and social and environmental experience, without and with playback experiments. It applies state-of-the-art methods for reducing the dimensionality of the data and finding patterns of correlation between different kinds of variables (factor analysis, K-means). The strength of the evidence, collated from a large number of trials with many controls, leads to the conclusion that the traditionally assumed communication function of chirps may be secondary to its role in environmental assessment and exploration that takes social context into account. Based on their extensive analyses, the authors suggest that chirps are mainly used as probes that help detect beats caused by other fish as well as objects.

Strengths:

The work is based on completely novel recordings using interaction chambers. The amount of new data and associated analyses is simply staggering, and yet, well organized in presentation. The study further evaluates the electric field strength around a fish (via modelling with the boundary element method) and how its decay parallels the chirp rate, thereby relating the above variables to electric field geometry. The BEM modelling also convincingly predicts how the electric image of a receiver conspecific on a sending fish is enhanced by a chirp.

The main conclusions are that the lack of any significant behavioural correlates for chirping, and the lack of temporal patterning in chirp time series, cast doubt on a primary communication goal for most chirps. Rather, the key determinants of chirping are the difference in frequency between two interacting conspecifics as well as individual subjects' environmental and social experience. The paper concludes that there is a lack of evidence for stereotyped temporal patterning of chirp time series, as well as of sender-receiver chirp transitions beyond the known increase in chirp frequency during an interaction. The authors carefully submit that the new putative echolocation function of chirps is not mutually exclusive with a possible communication function.

These conclusions by themselves will be very useful to the field. They will also allow scientists working on other "communication" systems to perhaps reconsider and expand the goals of the probes used in those senses. A lot of data are summarized in this paper, with thorough referencing to past work.

The alternative hypotheses that arise from the work are that chirps are mainly used as environmental probes for better beat detection and processing and object localization, and in this sense are self-directed signals. This led to their prediction that environmental complexity ("clutter") should increase chirp rate, which is fact was revealed by their new experiments. The authors also argue that waveform EODs have less power across high spatial frequencies compared to pulse-type fish, with a resulting relatively impoverished power of resolution. Chirping in wave-type fish could temporarily compensate for the lower frequency resolution while still being able to resolve EOD perturbations with a good temporal definition (which pulse-type fish lack due to low pulse rates).

The authors also advance the interesting idea that the sinusoidal frequency modulations caused by chirps are the electric fish's solution to the minute (and undetectable by neural wetware) echo-delays available to it, due to the propagation of electric fields at the speed of light in water. The paper provides a number of experimental avenues to pursue in order to validate the non-communication role of chirps.

---

## [Author Response]

The following is the authors’ response to the previous reviews.

**eLife Assessment**
This study addresses a question in sensory ethology and active sensing in particular. It links the production of a specific signal - electrosensory chirps - to various contexts and conditions to argue that the main function is to enhance conspecific localization rather than communication as previously believed. The study provides a lot of valuable data, but the methods section is incomplete making it difficult to evaluate the claims.

We have now added to the methods a new paragraph describing in better detail the analysis done to prepare the data used in figure 7. The figure itself has been substantially changed: we now show EOD fields and electric images using voltage, instead of current and we have better illustrated the comparisons between chirps and beats using statistical analysis.

Eventually, we are equally grateful to all Reviewers for the constructive criticism and for the time spent in evaluating our manuscript. It certainly helped to improve both the quality of the data presented as well as the readability of the text.

**Public Reviews:**

**Reviewer #1 (Public Review):**
The authors investigate the role of chirping in a species of weakly electric fish. They subject the fish to various scenarios and correlate the production of chirps with many different factors. They find major correlations between the background beat signals (continuously present during any social interactions) or some aspects of social and environmental conditions with the propensity to produce different types of chirps. By analyzing more specifically different aspects of these correlations they conclude that chirping patterns are related to navigation purposes and the need to localize the source of the beat signal (i.e. the location of the conspecific).The study provides a wealth of interesting observations of behavior and much of this data constitutes a useful dataset to document the patterns of social interactions in these fish. Some data, in particular the high propensity to chirp in cluttered environments, raises interesting questions. Their main hypothesis is a useful addition to the debate on the function of these chirps and is worth being considered and explored further.After the initial reviewers' comments, the authors performed a welcome revision of the way the results are presented. Overall the study has been improved by the revision. However, one piece of new data is perplexing to me. The new figure 7 presents the results of a model analysis of the strength of the EI caused by a second fish to localize when the focal fish is chirping. From my understanding of this type of model, EOD frequency is not a parameter in the model since it evaluates the strength of the field at a given point in time. Therefore the only thing that matters is the phase relationship and strength of the EOD. Assuming that the second fish's EOD is kept constant and the phase relationship is also the same, the only difference during a chirp that could affect the result of the calculation is the potential decrease in EOD amplitude during the chirp. It is indeed logical that if the focal fish decreased its EOD amplitude the target fish's EOD becomes relatively stronger. Where things are harder to understand is why the different types of chirps (e.g. type 1 vs type 2) lead to the same increase in signal even though they are typically associated with different levels of amplitude modulations. Also, it is hard to imagine that a type 2 chirp that is barely associated with any decrease in EOD amplitude (0-10% maybe), would cause a doubling of the EI strength. There might be something I don't understand but the authors should provide a lot more details on how this result is obtained and convince us that it makes sense.

We hope we have now resolved the Reviewer’s concerns by applying major edits to Figure 7. We now use voltage - not current - to quantify the impact of chirps on electric images. The effect of chirps is here estimated using the integral of the beat AM, as a broad measure of the potential effects chirping may have on electroreceptors. We underline in the text that this analysis does not represent proof for any type of processing occurring in the fish brain, but we only express in hypothetical terms that - based on the beat perturbations measured - additional spatial information may potentially be available in electric images, as a consequence of chirping. Whether the fish uses this information, or not, needs to be assessed through electrophysiology in future studies.

Finally, the reviewer is concerned about this sentence in the rebuttal - "The methods section has been edited to clarify the approach (not yet)". This section is unfinished, which suggests that it is difficult to explain the modeling results from a logical point of view. Thus the reviewer's major concern from the previous review remains unresolved. To summarize, the model calculates field strengths at an instant in time and integrates over time with a 500 ms window. This window is 10 times longer than the small chirps, while the longer chirps cover a much larger proportion of the window. Yet, the small chirps have a bigger impact on discriminability than the longer chirps. The authors should attempt to explain this seemingly contradictory result. This remains a major issue because this analysis was the most direct evidence that chirping could impact localization accuracy.

We added a new method section describing the new figure and hopefully it is explaining more clearly how the effect of chirps is calculated. Since most p-units are affected by the beat cyclic AMs, any change on the electric image caused by a chirp will result in changes in transcutaneous voltage - i.e. the voltage measurable at the receptor level. Overall, this added analysis is not a central point of the manuscript, it is part of an attempt to hint to physiological mechanisms implied which cannot be explored in the current study. We do not mean to propose that these estimates represent alternatives to electrophysiological recordings, rather theoretical evidences which could in fact support this type of investigation.

**Reviewer #2 (Public Review)**:Studying Apteronotus leptorhynchus (the weakly electric brown ghost knifefish), the authors provide evidence that 'chirps' (brief modulations in the frequency and amplitude of the ongoing wave-like electric signal) function in active sensing (specifically homeoactive sensing) rather than communication. Chirping is a behavior that has been well studied, including numerous studies on the sensory coding of chirps and the neural mechanisms for chirp generation. Chirps are largely thought to function in communication behavior, so this alternative function is a very exciting possibility that should have a great impact on the field.The authors provide convincing evidence that chirps may function in homeoactive sensing. In particular, the evidence showing increased chirping in more cluttered environments and a relationship between chirping and movement are especially strong and suggestive. Their evidence arguing against a role for chirps in communication is not as strong. However, based on an extensive review of the literature, the authors conclude, I think fairly, that the evidence arguing in favor of a communication function is limited and inconclusive. Thus, the real strength of this study is not that it conclusively refutes the communication hypothesis, but that it calls this hypothesis into question while also providing compelling evidence in favor of an alternative function.In summary, although the evidence against a role for chirps in communication is not as strong as the evidence for a role in active sensing, this study presents very interesting data that is sure to stimulate discussion and follow-up studies. The authors acknowledge that chirps could function as both a communication and homeactive sensing signal, and the language arguing against a communication function is appropriately measured. A given electrical behavior could serve both communication and homeoactive sensing. I suspect this is quite common in electric fish (not just in gymnotiforms such as the species studied here, but also in the distantly related mormyrids), and perhaps in other actively sensing species such as echolocating animals.

We are grateful to the Reviewer for the kind assessment.

**Reviewer #3 (Public Review):**
Summary:This important paper provides the best-to-date characterization of chirping in weakly electric fish using a large number of variables. These include environment (free vs divided fish, with or without clutter), breeding state, gender, intruder vs resident, social status, locomotion state and social and environmental experience, without and with playback experiments. It applies state-of-the-art methods for reducing the dimensionality of the data and finding patterns of correlation between different kinds of variables (factor analysis, K-means). The strength of the evidence, collated from a large number of trials with many controls, leads to the conclusion that the traditionally assumed communication function of chirps may be secondary to its role in environmental assessment and exploration that takes social context into account. Based on their extensive analyses, the authors suggest that chirps are mainly used as probes that help detect beats caused by other fish as well as objects.Strengths:The work is based on completely novel recordings using interaction chambers. The amount of new data and associated analyses is simply staggering, and yet, well organized in presentation. The study further evaluates the electric field strength around a fish (via modelling with the boundary element method) and how its decay parallels the chirp rate, thereby relating the above variables to electric field geometry. The BEM modelling also convincingly predicts how the electric image of a receiver conspecific on a sending fish is enhanced by a chirp.The main conclusions are that the lack of any significant behavioural correlates for chirping, and the lack of temporal patterning in chirp time series, cast doubt on a primary communication goal for most chirps. Rather, the key determinants of chirping are the difference in frequency between two interacting conspecifics as well as individual subjects' environmental and social experience. The paper concludes that there is a lack of evidence for stereotyped temporal patterning of chirp time series, as well as of sender-receiver chirp transitions beyond the known increase in chirp frequency during an interaction. The authors carefully submit that the new putative echolocation function of chirps is not mutually exclusive with a possible communication function.These conclusions by themselves will be very useful to the field. They will also allow scientists working on other "communication" systems to perhaps reconsider and expand the goals of the probes used in those senses. A lot of data are summarized in this paper, with thorough referencing to past work.The alternative hypotheses that arise from the work are that chirps are mainly used as environmental probes for better beat detection and processing and object localization, and in this sense are self-directed signals. This led to their prediction that environmental complexity ("clutter") should increase chirp rate, which is fact was revealed by their new experiments. The authors also argue that waveform EODs have less power across high spatial frequencies compared to pulse-type fish, with a resulting relatively impoverished power of resolution. Chirping in wave-type fish could temporarily compensate for the lower frequency resolution while still being able to resolve EOD perturbations with a good temporal definition (which pulse-type fish lack due to low pulse rates).The authors also advance the interesting idea that the sinusoidal frequency modulations caused by chirps are the electric fish's solution to the minute (and undetectable by neural wetware) echo-delays available to it, due to the propagation of electric fields at the speed of light in water. The paper provides a number of experimental avenues to pursue in order to validate the non-communication role of chirps.

We are grateful to the Reviewer for the kind assessment.